# The −10 region adjacent to open reading frames is a common expression pattern in *Deinococcus-Thermus*

Shitong Zhong[1],*, Linjia Wang[1],*, Shuang Song[1], Liangyan Wang[1], Yuejin Hua[1,2], Huizhi Lu[1]

Gene transcription and translation are fundamental processes that underpin the vitality of living organisms. In bacteria, the classic promoter elements for transcription include the −35 region and the −10 region, whereas the ribosome binding site (RBS) is essential for translation. Our investigation of the upstream sequences of ORFs in *Deinococcus radiodurans* has confirmed a −10 region-like motif, which appears as 5′-TANNNT-3′ (−10-motif). Through rigorous experimental validation, we have established that this −10-motif functions as the classical −10 region of the promoter during transcription, responsible for initiating transcription of leaderless genes (mRNA products that do not have or only have a very short 5′-UTR), and exhibits specific spacing requirements relative to the ORF. Moreover, our findings suggest that sequences containing −10-motif can serve as promoters for *D. radiodurans* and introduction of the −35 region at appropriate positions can significantly enhance transcription levels. Genomic analysis revealed that the −10-motif located a few base pairs (bp) upstream of ORFs is common within the *Deinococcus-Thermus* (*D-T*) phylum (also known as *Deinococcota*) and responsible for the transcriptional initiation of leaderless genes. Given these insights, we propose that the −10 region adjacent to the ORF for gene expression via leaderless mRNA constitutes an important mode of expression in the *D-T* phylum, which may contribute to the extraordinary environmental adaptability observed in this group of microorganisms.

## Introduction

Gene transcription and translation are pivotal in the life cycle of an organism, governing the conversion of genetic information into functional proteins. Central to these processes are the promoter and the ribosome binding site (RBS), which are strategically positioned upstream of an ORF. RNA polymerase (RNAP) recognizes and binds to the promoter, thereby initiating transcription and resulting in the synthesis of mRNA. After transcription, the ribosome identifies the RBS on the mRNA molecule, marking the initiation for translation and ultimately leading to the production of polypeptides.

Prokaryotic promoter research has largely been centered on *Escherichia coli*. In 1974, Schaller's work elucidated the characteristics of the RNAP binding site (Schaller et al, 1975). Pribnow's subsequent work identified the sequence 5′-TATPuATG-3′ as critical for the formation of a stable complex between the promoter and RNAP (Pribnow 1975), which contributed to the nomenclature of the −10 region as the Pribnow box. As research advanced, Takanami and colleagues identified a sequence ~35 base pairs upstream of the transcription initiation site that is crucial for transcription recognition (Takanami et al, 1976). In *E. coli*, sequence alignment upstream of the transcription start site has revealed that the −35 region typically matches 5′-TTGACA-3′, and the −10 region aligns with 5′-TATAAT-3′ (Hawley & McClure 1983; Harley & Reynolds 1987). Extensive research has subsequently confirmed that the sigma factor (RpoD) plays a key role in interacting with the −35 and −10 regions, thereby facilitating the accurate initiation of transcription (Simpson 1979; Morett & Buck 1989; Bowers & Dombroski 1999).

In the exploration of the RBS, initial theories posited that translation initiation was dependent on the recognition of specific nucleotide sequences or the identification of secondary or tertiary structures in the mRNA (Hindley & Staples 1969; Steitz 1969). However, the discovery of the 3′-terminal sequence of 16S rRNA and its complementarity to a sequence upstream of the initiation codon implied that ribosomes recognize the initiation site through base pairing with the 5′-UTR of the mRNA (Shine & Dalgarno 1974; Steitz & Jakes 1975). Subsequent experimental validations confirmed that the ribosome recognition sequence on mRNA correlates with the 5′-CCUCCU-3′ sequence at the end of 16S rRNA (Hui & de Boer 1987; Jacob et al, 1987). This led to the identification of the classical RBS as the sequence 5′-AGGAGG-3′ or 5′-GGAGG-3′, which is commonly referred to as the Shine–Dalgarno (SD) sequence.

The field of transcription and translation research continues to yield new insights that mRNAs lacking or possessing very short 5′-UTRs can still be normally translated (Shean & Gottesman 1992).

---

[1]MOE Key Laboratory of Biosystems Homeostasis and Protection, Institute of Biophysics, College of Life Sciences, Zhejiang University, Hangzhou, China    [2]Cancer Center, Zhejiang University, Hangzhou, China

Correspondence: yjhua@zju.edu.cn; huizhilu@zju.edu.cn
*Shitong Zhong and Linjia Wang contributed equally to this work

These mRNAs, referred to as leaderless mRNAs, are widely distributed among bacteria, fungi, and archaea (Moll et al, 2002). This interesting phenomenon indicates that the upstream regions of these ORFs contain solely the promoter without RBS. Furthermore, studies have revealed a variety of promoter motifs in bacterial genomes (Ponnambalam et al, 1986; Keilty & Rosenberg 1987; Morett & Buck 1989; Gleghorn et al, 2008). The expansion of genomic databases has enabled bioinformatics analyses showing that sequences upstream of ORFs reveal a diversity in base composition across different bacterial species (Omotajo et al, 2015; Nakagawa et al, 2017). This suggests that the large amount of information contained in the DNA sequence upstream of the ORFs requires further investigation into its possible connection to the promoter or RBS. Given the intricacy of the upstream ORF sequences, it may be necessary to reconsider the definition/ constitution of promoters (Mejía-Almonte et al, 2020).

Promoter research within the *Deinococcus-Thermus* (*D-T*) phylum has been largely centered on the genus *Thermus*, with a focus on understanding the role of transcriptional motifs (Feklistov et al, 2006; Sevostyanova et al, 2007; Miropolskaya et al, 2018) and their interactions with RNAP (Zhang et al, 2012; Weixlbaumer et al, 2013; Bae et al, 2015), as well as comparisons of promoter–RNAP interaction characters with *E. coli* (Mekler et al, 2012; Miropolskaya et al, 2012). Despite some efforts to summarize known promoter sequences (Sevostyanova et al, 2007), a comprehensive genomic overview of promoter organization within this phylum remains absent. In contrast, a research on *Deinococcus* promoters has been more piecemeal, with studies failing to converge on a unified set of distinct patterns (Chen et al, 2019). Nevertheless, analysis of the sequences upstream of ORFs in *Deinococcus* has uncovered a motif (5′-TANNNT-3′) that closely resembles the classical –10 region in *E. coli*, suggesting a potential role of this motif in transcription (Zheng et al, 2011; de Groot et al, 2014).

Consistent with a previous study, our analysis of the DNA sequence upstream of all ORFs in *Deinococcus radiodurans* (*D. radiodurans*) also revealed the conserved motif with the sequence 5′-TANNNT-3′, designated as the –10-motif. This motif displays a high degree of conservation within *Deinococcus*, suggesting its crucial role in gene expression. Through a series of experiments, we have demonstrated that the –10-motif is indeed associated with transcription and plays a classical –10 region role in RNAP recognition. Mutations at the conserved sites of the –10-motif led to abnormal gene expression. Our findings suggest that approximately one third of genes in *D. radiodurans* are transcribed as leaderless mRNA, raising concerns about the accuracy of some gene annotations. Notably, when a –10-motif appears at an appropriate position on a sequence, it can function as a promoter sequence in *D. radiodurans*. In addition, the presence of the –35 region can significantly enhance transcriptional expression levels. What's more, our sequence alignment has revealed that the –10-motif is a widespread feature across various species within the *D-T* phylum, and the function assays demonstrated a conserved role of –10-motif in transcriptional regulation. Thus, we posit that the –10-motif is a significant promoter pattern in the *D-T* phylum.

In the classical expression model primarily studied in *E. coli*, gene transcription relies on promoters containing –35 and –10 regions to produce mRNA with 5′-UTR; and subsequent translation depends on the RBS located within this 5′-UTR. In contrast, our research reveals that a significant number of genes in the *D-T* phylum use transcription relying on a –10 region immediately upstream of the ORF to produce leaderless mRNA for protein expression. This pattern is markedly different from the classical expression model.

## Results

### –10-motif forms a nonclassical expression pattern in *D. radiodurans*

Through DNA sequence analysis on sequences 20 bp upstream of *D. radiodurans* ORFs using MEME software (Bailey et al, 2015), two primary motifs were identified (Fig 1A), consistent with findings in *Deinococcus deserti* (de Groot et al, 2014). One motif (–10-motif) presents in the form of TANNNT, similar to the classical –10 region of promoter, whereas the other (RBS-motif) presents in the form of GGAG, similar to classical RBS sequences. These two predicted motifs, although similar in sequence to known functional sequences, still require further validation for their specific functions. Because the RBS-motif not only shares similar features in sequence with classical RBS sequences but also appears a few bp upstream of the ORF, matching the positional characteristics of classical RBS, we infer that it is the RBS sequence in *D. radiodurans*. In contrast, –10-motif occupies a position similar to that of the RBS, but exhibits a distinct sequence. It is similar to the –10 region of classical promoters and is most frequently found in the form of 5′-TACACT-3′ (Fig S1A). Notably, the –10-motif predominantly occurs ~6–7 bp upstream of the ORF (Fig S1B and C), present in nearly one-third of the *D. radiodurans* genes (Fig 1A), and does not show a bias toward specific pathways (Fig S1D), which is consistent with previous findings (de Groot et al, 2014).

Notably, the –10-motif is present and highly conserved across various species within the *Deinococcus* genus, with the first T, second A, and sixth T remaining nearly invariant (Fig S2A–E). Interestingly, some –10-motifs appear within the reading frame or far upstream, which suggest inaccuracies in previous reading frame annotation, such as the –10-motif of the *pprI* (also known as *irrE*) (Fig S2E), which has been discussed by a previous study (Ludanyi et al, 2014). With the –10-motif ahead of the ORF, the transcript is assumed to lack 5′-UTR, which was identified as leaderless mRNA by previous studies (Zheng et al, 2011; de Groot et al, 2014).

In *D. radiodurans*, some proteins such as GroES and DdrO are expressed through the traditional transcription and translation pathways, using classical promoter sequences that are followed by RBS sequences (Fig 1B). To determine whether there is any regulation preference of –10-motifs adjacent to ORFs versus classical promoters, we performed enrichment analysis on the transcriptome data of *D. radiodurans* (SRX2731648) to observe RNA transcription levels near the ORF. Analysis of RNA transcription levels reveals a substantial elevation following the classical promoter's –10 region (Fig 1C and D), consistent with the known function of promoter in initiating transcription. Similarly, we observed a notable increase in transcription levels following the –10-motif (Figs 1E and F and S3A–F), suggesting that the –10-motif could

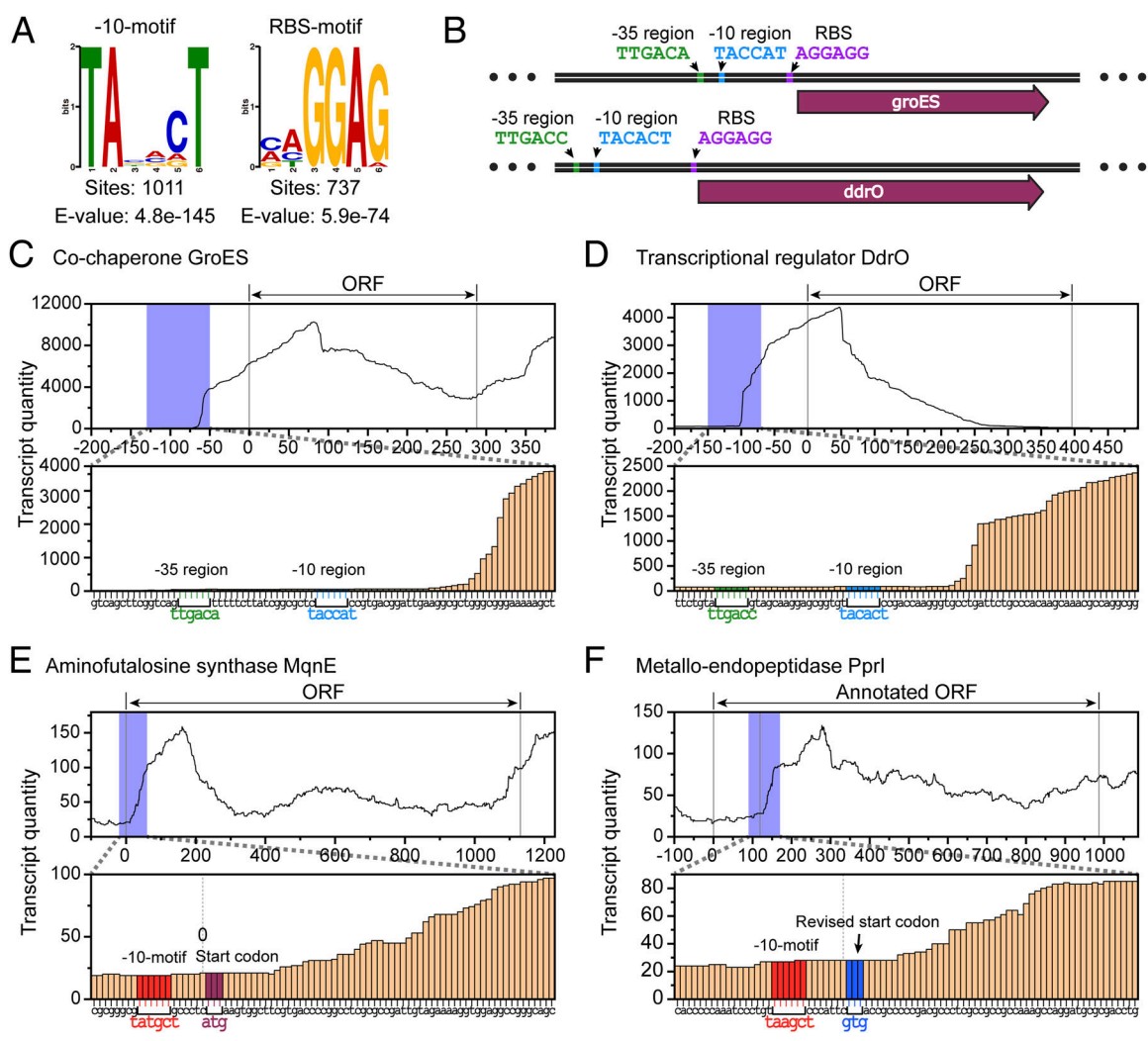

**Figure 1. Performance of motifs upstream of the ORFs and RNA transcription in *D. radiodurans*.**
**(A)** Two motifs were identified by scanning the 20-bp sequence upstream of the ORFs (the total number is 3,146) in the *D. radiodurans* genome. The left −10-motif closely resembles the −10 region of a classical promoter, whereas the right ribosome binding site (RBS)-motif corresponds to a classical RBS sequence. **(B)** Classical transcription and translation patterns in *D. radiodurans*, exemplified by *groES* and *ddrO*. The −35 region, −10 region, and RBS are highlighted in green, blue, and purple, respectively. **(C, D, E, F)** RNA transcription statistics of genome sequences. **(C, D)** Transcriptome data reveal that following the promoters (highlighted in green and blue) of GroES (C) and DdrO (D), a clear elevation in transcription levels of related sequences is observed, aligning with the expected function of promoters. **(E, F)** Similarly, after the −10-motif (highlighted in red) of MqnE (E) and PprI (F), a notable increase in transcription levels is evident, suggesting that the function of −10-motif parallels that of a promoter. (The bar chart below is created by enlarging the blue section from the line graph above. The annotated start codon of the ORF is indicated in dark purple, whereas the revised start codon is marked in blue).

be a regulatory element like classical promoters, which controls gene transcription in *D. radiodurans*.

## −10-motif is important for gene expression

Previous sequence analysis has shown that many species have their ORFs closely followed by RBSs upstream (Omotajo et al, 2015; Nakagawa et al, 2017), and therefore, it can be inferred that the classical promoter −10 regions for these ORFs should be located dozens bp upstream. In contrast, the genomic location of −10-motif in *D. radiodurans* is very close to the ORF, which markedly differs from that of classical promoters, pointing out the possibility of the −10-motif acting as a RBS. To investigate the role of the −10-motif, we constructed a

reporter plasmid using the upstream sequence of aminofutalosine synthase *mqnE* (pMqnE) and the modified upstream sequence of metalloprotease *pprI* (pPprI), followed by *eGFP* as a reporter gene (Fig 2A). The upstream sequence of *groES* (pGroES) and *mCherry* was fused for background referencing, and the upstream sequence of *ddrO* (pDdrO) coupled with *eGFP* for control (Fig 2A). We introduced random mutation into the −10-motif of pMqnE and pPprI, or substituted them with a classical RBS sequence. In addition, random mutation was also introduced into the −10 region of pDdrO (Fig 2B). The constructed plasmids were transformed into *D. radiodurans* R1 strain, and the expression of eGFP protein was monitored using fluorescence microscopy. As well as the mutations, the substitution of −10-motif with classical RBS also led to the loss of green fluorescence, indicating the

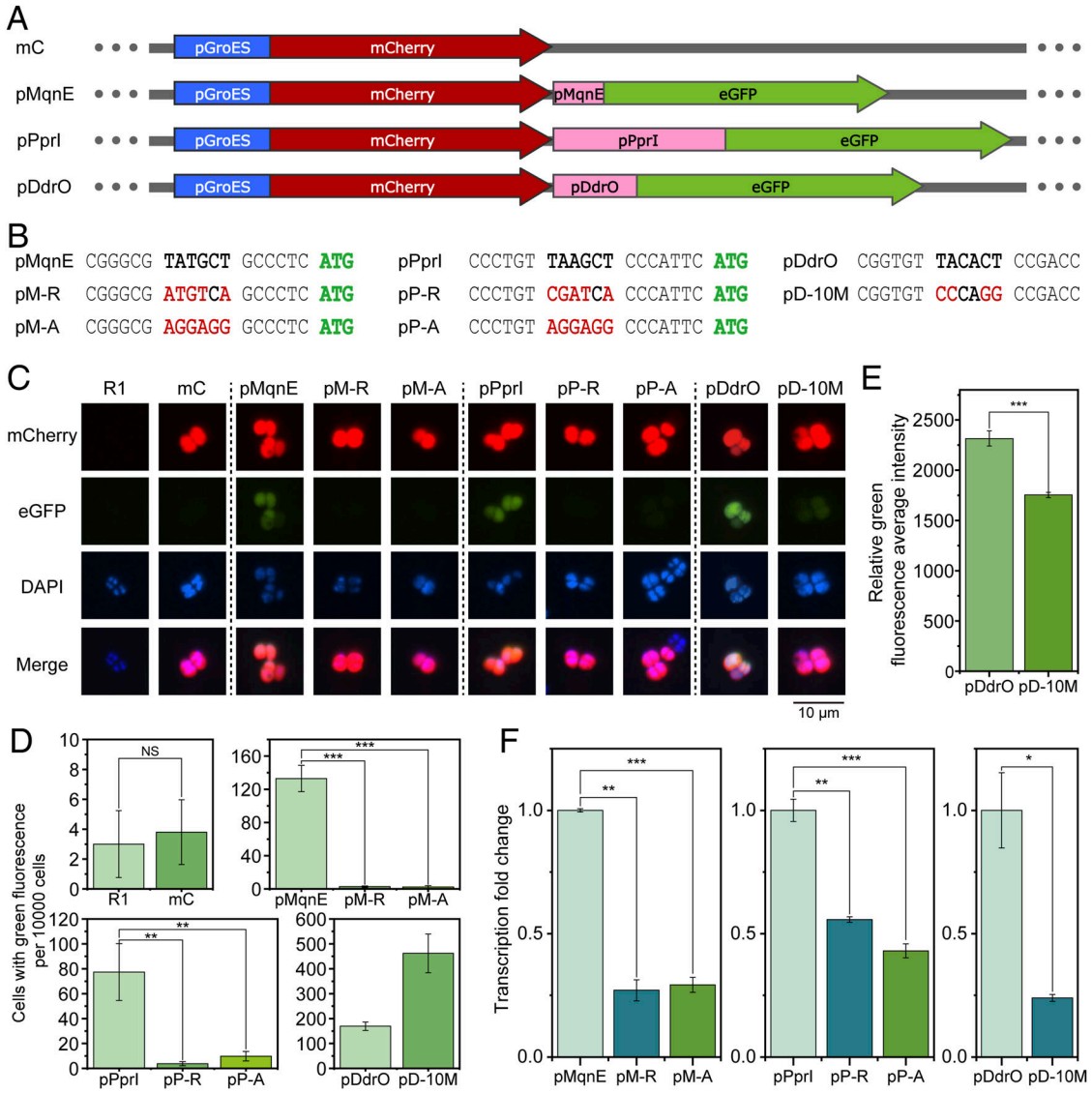

**Figure 2. Impact of −10-motif substitution on protein expression.**
**(A)** Construction of four fluorescent reporter vectors, where mCherry served as a reference linked to the upstream sequence of the *groES* (pGroES), and eGFP acted as an indicator linked to the upstream sequences of the *mqnE* (pMqnE), *pprI* (pPprI, new ORF start position was used), and *ddrO* (pDdrO). **(B)** Replacement of key sites in the pMqnE, pPprI, and pDdrO, in which the −10-motif of pMqnE and pPprI was replaced with either a random sequence (pM-R and pP-R) or a classical ribosome binding site (pM-A and pP-A), and the −10 region of the DdrO promoter was substituted with a random sequence (pD-10M). **(C)** Expression of fluorescent proteins mCherry and eGFP in *D. radiodurans*. Results showed that the replacement of the −10-motif in MqnE and PprI resulted in the disappearance of eGFP fluorescence, whereas the replacement of the −10 region in DdrO promoter resulted in the reduced eGFP fluorescence. **(D, E)** Green fluorescence expression captured by flow cytometry. **(F)** Transcription level of eGFP was assessed relative to the WT sequences, revealing that the substitution of key sites resulted in decreased transcription levels. NS: $P \geq 0.05$, *$P < 0.05$, **$P < 0.01$, ***$P < 0.001$, ****$P < 0.0001$.

different role of −10-motif compared with RBS. In contrast, mutation in the −10 region of pDdrO resulted only in a diminished fluorescence intensity, suggesting a different transcription mode between the −10-motif and classical −10/−35 promoter (Fig 2C). The fluorescent observations were further substantiated by cell flow cytometry data (Fig 2D and E). Using the transcription level of the mCherry protein as a baseline, we quantified the transcription level of eGFP through qPCR. The results indicate that occupation of the −10-motif by alternate sequences leads to a marked reduction in RNA levels, similar to the behavior observed in pDdrO with mutations in the −10 region (Fig 2F).

Consequently, we affirmed the role of −10-motif as a promoter. Nevertheless, the −10-motif necessitates a specific proximity to the initiation codon, whereas an excessively close distance would lead to aberrant expression (Fig S4A and B).

### The −10-motif functions as the classical −10 region

Previous research on the crystal structure of RNAP from *Thermus thermophilus* (PDB: 4G7H) has identified a crucial domain within RpoD that interacts with the −10 region. Given

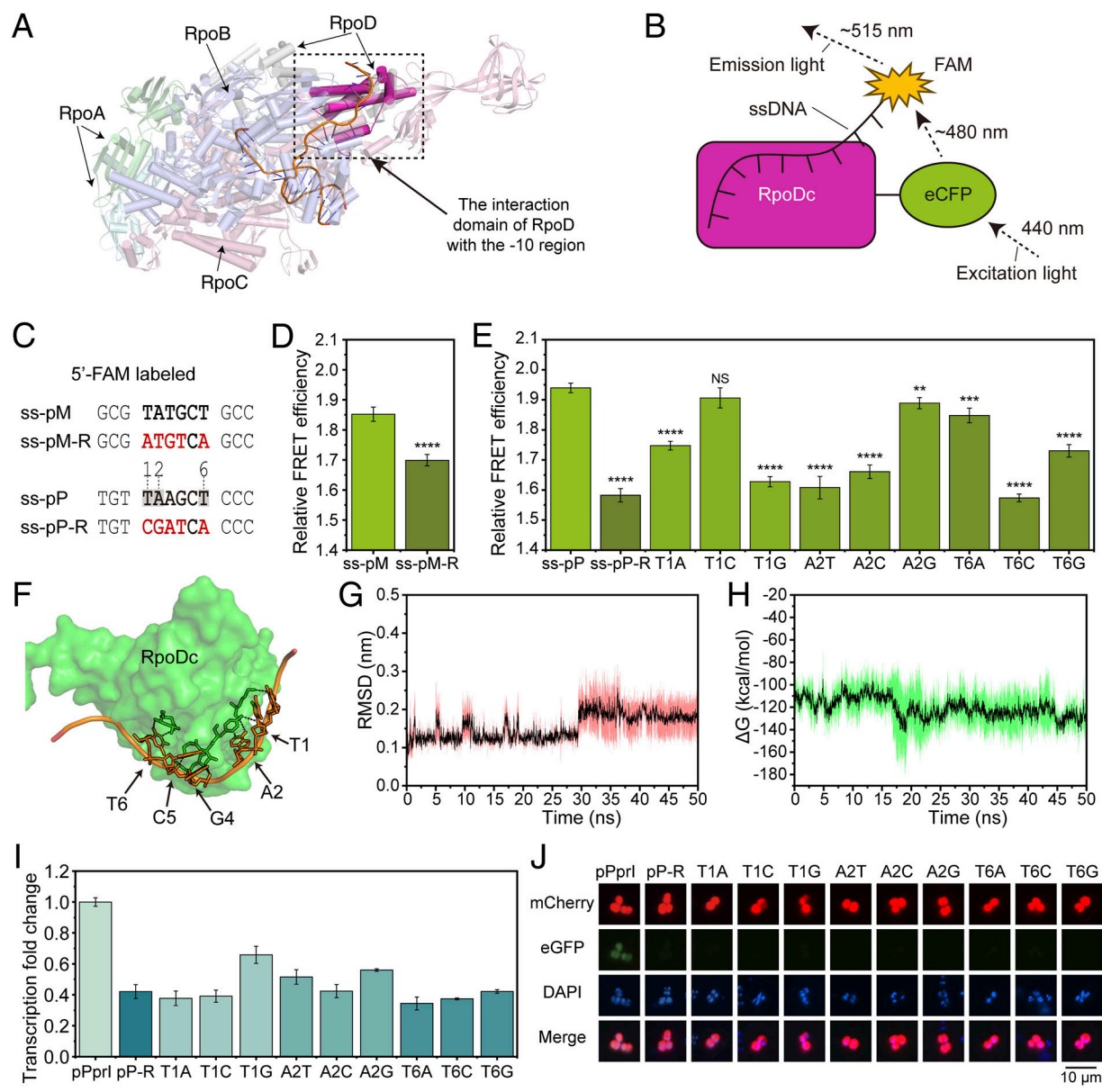

**Figure 3. Interactions between the −10-motif and RpoD's corresponding domain, and the effect of point mutations on interaction and protein expression.**
**(A)** Interaction between RNA polymerase and the −10 region. The crystal structure of RNA polymerase was from *T. thermophilus* (annotated in light color, PDB: 4G7H), and the truncated RpoD (RpoDc) from *D. radiodurans* was predicted by AlphaFold (indicated in purple). **(B)** Design diagram of fluorescence resonance energy transfer (FRET) experiments, in which the C terminus of RpoDc is conjugated to eCFP and ssDNA is tagged with 5′-FAM, enabling FRET upon molecular interaction. **(C)** Four ssDNAs labeled with 5′-FAM were designed based on −10-motifs in pMqnE and pPprI. The −10-motif of pPprI was modified to introduce point mutations at the first, second, and sixth nucleotide positions, respectively. **(D, E)** FRET results of four ssDNAs and ss-pP point mutations with RpoDc. **(F)** Interaction structure of RpoDc with −10-motif predicted by AlphaFold. Hydrogen bonds are indicated by dashed lines. The numbered bases belong to the −10-motif, with the exception of A2 and T6, where all other bases interact via the phosphate backbone. **(G, H)** Results from molecular dynamics simulations of the predicted structure: the root mean square deviation of the −10-motif (sequence: TAAGCT) (G) and the interaction free energy ΔG between protein and DNA (H). **(I, J)** Transcription-level alterations (I) and expression results (J) of eGFP in *D. radiodurans* after point mutations in the −10-motif of PprI. NS: *P* ≥ 0.05, *\*P* < 0.05, *\*\*P* < 0.01, *\*\*\*P* < 0.001, *\*\*\*\*P* < 0.0001.

that the predicted structure of *D. radiodurans*'s RpoD aligns well with this domain and the −10-motif closely resembles the classical −10 region, we hypothesized that −10-motif may interact with the RpoD protein (Fig 3A). To test the interaction between the −10-motif and RpoD, we constructed a truncated RpoD (RpoDc) and linked it to the eCFP protein at the C terminus via a flexible peptide linker. Meanwhile, we labeled a 5′-FAM tag

to the single-stranded DNA (ssDNA) containing the −10-motif (from pMqnE and pPprI), and assessed the interaction between −10-motif and RpoD through fluorescence resonance energy transfer (FRET) between FAM and eCFP (Fig 3B and C). Taking the intensity ratio of 515-nm emission light to 480-nm emission light as an indicator of FRET efficiency, we observed strong interaction between −10-motif and RpoD, whereas the mutated

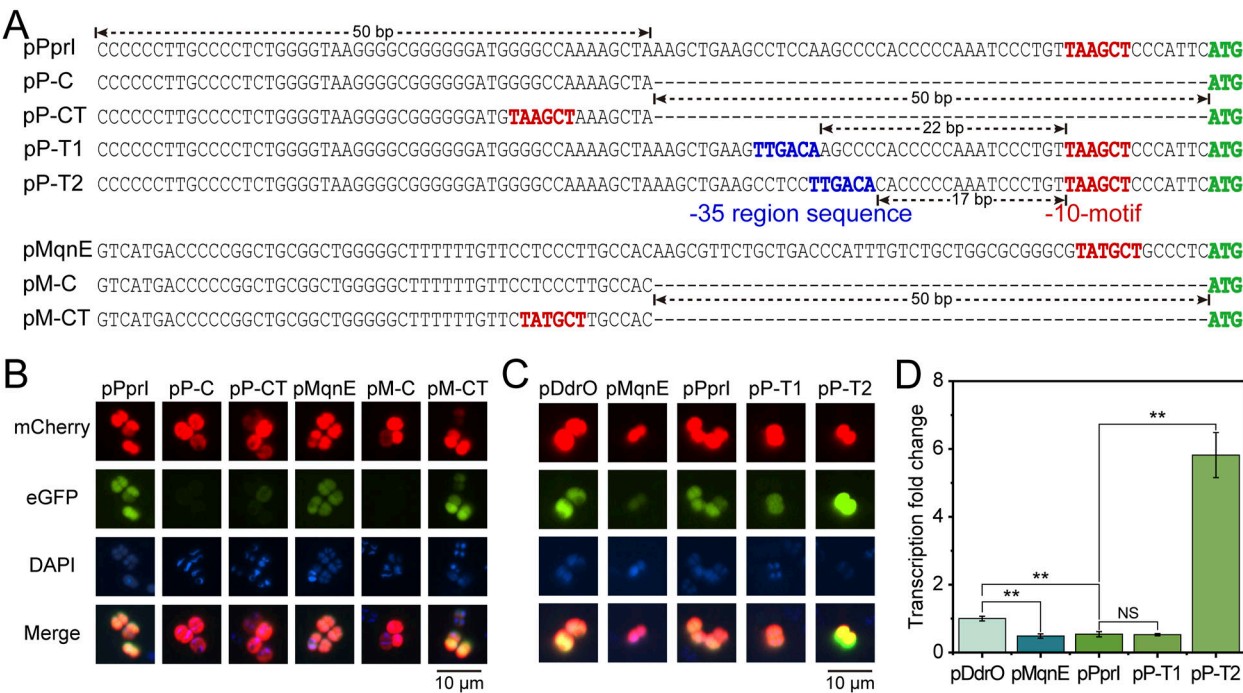

**Figure 4. Dependency of the −10-motif on other sequences.**
**(A)** pPprI and pMqnE were truncated by 50 bp (pP-C, pM-C), and the −10-motif was added to the truncated sequences (pP-CT, pM-CT). The −35 region sequences were introduced at 22 bp (pP-T1) and 17 bp (pP-T2) upstream of the −10-motif in pPprI, respectively. **(B)** Impact of a 50-bp truncation adjacent to the start codon and the restoration of the −10-motif on the expression of fluorescence. **(C, D)** Effects of −35 region sequence on fluorescence expression. NS: $P \geq 0.05$, $*P < 0.05$, $**P < 0.01$, $***P < 0.001$, $****P < 0.0001$.

−10-motif exhibited a significant decrease in FRET efficiency (Fig 3D and E), suggesting that the −10-motif interacts with the RpoD and indeed functions as the −10 region. We further used AlphaFold to predict the complex structure of RpoDc with −10-motif, exhibiting multihydrogen bonds and salt bridges participating in the interaction (Fig 3F). The binary model was then validated by molecular dynamics simulations, with the root mean square deviation (RMSD < 0.2 nm) of the −10-motif showing the stability of the binding between RpoDc and −10-motif (Fig 3G). The calculated interaction free energy ΔG further demonstrated the high affinity between the protein and DNA (Fig 3H).

Except for T1C, the substitution of bases at positions 1, 2, and 6 with any other bases decreased the interaction significantly (Fig 3E), emphasizing the participation of the bases in interacting with RpoD. Meanwhile, in vivo mutations of these bases significantly reduced the transcription levels of eGFP compared with the WT (Fig 3I), leading to the decreased expression of the reporter eGFP (Fig 3J), suggesting that these three bases are extremely important for gene transcription, as well as the interaction with RpoD.

### Sequences containing the −10-motif can serve as promoters

Notably, the upstream sequences of most −10-motif genes do not exhibit the presence of a classical −35 region. To test whether there exists a sequence similar to the −35 region to aid −10-motif in normal gene expression, we excised the 50-bp segment upstream of the start codon and reinserted the −10-motif at the new position (Fig 4A). Analysis of fluorescent protein expression revealed that the 50-bp truncation eliminated the normal expression of fluorescent proteins, which could be restored by reintroducing the −10-motif (Fig 4B). This indicates that the presence of a −10-motif in the sequence is sufficient to function as a promoter in *D. radiodurans*, and also suggests that the RNAP in *D. radiodurans* has weak dependence on sequences other than the −10-motif. However, variations in fluorescence intensity upon reintroduction imply that the sequences adjacent to the −10-motif still have an impact on transcription strength (Fig 4B).

Generally, classical promoter in organisms like *E. coli* comprises the −10 region and the −35 region, both of which contribute to the recognition by sigma factor (Zuo & Steitz 2015). Through transcriptional and translational levels, we have demonstrated that the −10-motif upstream of ORFs functions as a classical −10 region in *D. radiodurans*. Furthermore, simply introducing the −10-motif sequence alone is sufficient to enable a segment of DNA to be recognized by RpoD (sigma factor) and function as a promoter (Fig 4A and B). Because both the −35 region and −10 region are required in classical promoters, it is possible that adding a −35 region sequence upstream of the −10-motif in *D. radiodurans* might stabilize or assist in protein expression. To evaluate the impact of the −35 region on the −10-motif, we integrated the −35 region into pPprI (followed by the eGFP gene) to assess its effect on fluorescence expression (Fig 4C and D). Compared with pDdrO, pMqnE and pPprI led to markedly lower transcription levels (Fig 4D). However, with the −35 region

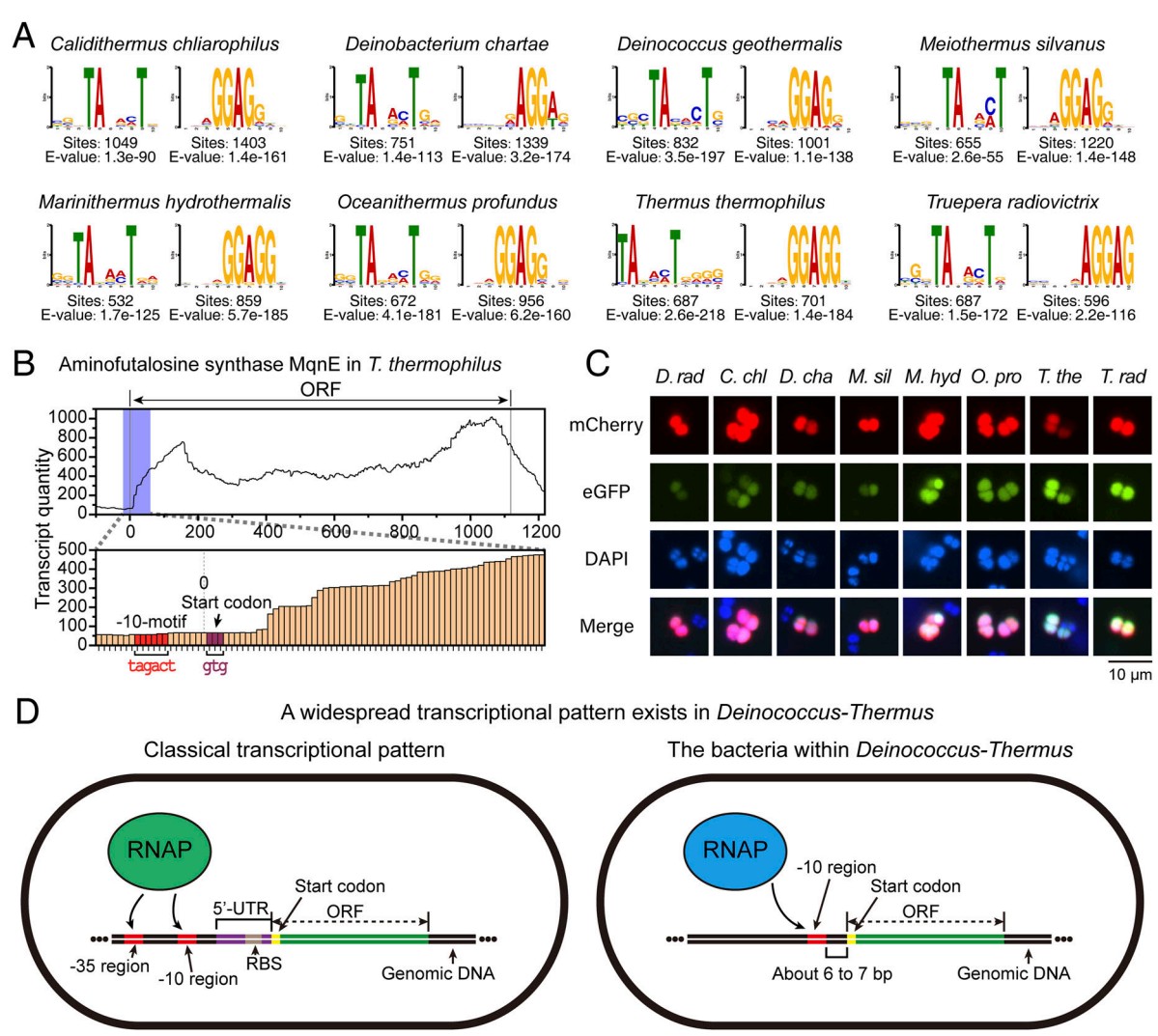

**Figure 5. Distribution and performance of −10-motif across *D-T* phylum.**
**(A)** Results of motif discovery within the upstream sequences of ORFs across various species within the *D-T* phylum. **(B)** Transcription-level statistics for the *mqnE* gene in *T. thermophilus* indicating a marked increase in transcription subsequent to the −10-motif. **(C)** Outcomes of substituting the 50-bp sequence proximal to the start codon in the *mqnE* ORF upstream sequence with the corresponding 50-bp sequence from other *D-T* phylum species. These sequences enabled the normal expression of eGFP in *D. radiodurans*, although there might be differences in fluorescence intensity. **(D)** Widespread transcriptional pattern in the *D-T* phylum.

positioned appropriately upstream (17 bp) of the −10-motif (pP-T2), the promoter significantly enhanced gene transcription and expression levels, even for the variants of the −10-motif (Figs 4C and D and S5B and C). When the −35 region is positioned inappropriately upstream (22 bp) of the −10-motif (pP-T1), the enhancement was gone (Figs 4C and D and S5A). In fact, existing studies have shown that the optimal distance between the −35 region and the −10 region is 17 ± 1 bp in *E. coli* (Aoyama et al, 1983) and *B. subtilis* (Yeh et al, 2011), which is consistent with our results.

### Ubiquity and functional consistency of the −10-motif in the *D-T* phylum

We also analyzed the upstream motifs of ORFs in species within the *D-T* phylum (also known as *Deinococcota*) and observed that the results are similar to those in *D. radiodurans*, with the clear presence

of the −10-motif and RBS-motif (Figs 5A and S6). Furthermore, we conducted an analysis of upstream sequences of ORFs in bacterial species across the entire domain, where the predominant motifs identified were again the −10-motif and RBS-motif (Fig S7A). Interestingly, all analyzed species within the *D-T* phylum exhibited the presence of the −10-motif (Fig S7B), with the average proportion of ORFs carrying the −10-motif reaching ~30% (Fig S7C), making it the most prominent among all phyla studied.

Taking *T. thermophilus* as an example, the transcription levels elevate following the −10-motif, which is similar to *D. radiodurans*, suggesting that the −10-motif across the *D-T* phylum uniformly acts as a promoter sequence (Figs 5B and S8A–F). Besides, the substitution of promoter sequence with 50-bp sequences upstream of the ORFs from other species before the eGFP-encoded sequence led to the normal expression of eGFP in *D. radiodurans* (Figs 5C and S9), demonstrating the universality of the −10-motif in the *D-T* phylum. In

addition, high resemblance in sequence alignment of the RpoD protein also suggested the similar interaction between −10-motif and RpoD (sigma factor) throughout the *D-T* phylum (Fig S10).

# Discussion

Sequences upstream of the ORFs are significant for normal transcription and translation of genes. In prokaryotes, the classical promoter for the recognition and binding of RNAP consists of the −35 region and −10 region. After transcription, the resulting mRNA typically contains 5′-UTR, where the RBS, related to translation, is located upstream of the start codon. However, researchers have identified leaderless mRNAs lacking or possessing very short 5′-UTRs, which can still be normally translated. Studies suggest that the initiation of leaderless mRNA translation is triggered by the interaction between the 70S particle and the start codon (O'Donnell & Janssen 2002; Udagawa et al, 2004; Beck et al, 2016), which is different from the conventional translation process that relies on initiation factors (Cummings & Hershey 1994; Tedin et al, 1999; Yamamoto et al, 2016).

This article reveals that apart from the RBS-motif, a conserved motif (−10-motif) similar to the classical −10 region is located ahead of the start codon for almost one-third ORFs in *D. radiodurans*, functioning as −10 region of classical promoter in transcription and resulting in disappeared or shortened 5′-UTR for the following leaderless mRNA.

Because the −10-motif is located ahead of the start codon like classical RBS, we designed experiments to investigate whether it functions like RBS. The results turned out that the −10-motif cannot be replaced by a classical RBS sequence, and the interaction between −10-motif and RpoD (sigma factor) further indicated that the −10-motif acts like −10 region of the classical promoter. We further confirmed that the 1/2/6 bases of the −10-motif are irreplaceable, consistent with structural findings of RpoD interacting with the −10 region (Feklistov & Darst 2011; Zhang et al, 2012; Zuo & Steitz 2015), confirming the −10-motif is a −10 region. Our experiments also demonstrated that in *Deinococcus*, a sequence with the −10-motif introduced at an appropriate position can function as a promoter. In contrast, different to that in *Deinococcus*, the introduction of −10-motif–containing plasmids constructed above into *E. coli* does not achieve eGFP expression, suggesting the disparate activity of RpoD in other species (Fig S11A and B). Meanwhile, free energy calculations also indicated that RpoD from *E. coli* has a weaker binding affinity to the −10 region (−10-motif) compared with *D. radiodurans* and *T. thermophilus* (Fig S12A–E). Besides, the diminished expression restored after the addition of −35 region into the proper position before the −10-motif (Fig S11A and B), confirming that RpoD from *E. coli* requires both the −10 and −35 region of the classical promoter to complete its mission of transcription initiation. However, for the leaderless genes in *Deinococcus*, the addition of the appropriately positioned −35 region led to enhancement of gene expression levels. In addition, −10-motifs appearing in different sequences exhibit varying expression strengths, and we speculate that this might be influenced by their adjacent sequences, which has also been discovered in *Thermus aquaticus* (Feklistov et al, 2006). Furthermore, the binding motifs of some transcriptional regulators, such as DdrO

binding site characterized by previous studies (Makarova et al, 2007; Wang et al, 2015), may overlap with the −10-motif or appear nearby (Fig S13), thereby regulating expression strength.

ORFs were used to be characterized through the identification of the start codon. However, in *Deinococcus*, proteins such as PprI, LspA, MurD, and RecF, which are encoded after the −10-motif, exhibit disordered peptide chains at the N terminus in the predicted or crystal structures (Figs S2E and S14A–C). The peptide chains float around the periphery of the tight tertiary structure, suggesting its possible nonfunctionality and the incorrect annotation for previous leaderless genes. The enzymatic activity of PprI without the peptide was intact (Lu et al, 2019, 2024), further confirming the annotation error. Thus, the identification of the −10-motif as the promoter also enabled us to reannotate the leaderless genes.

In addition, we found that the −10-motif is not only prevalent in *Deinococcus* but also observed across various species within the *D-T* phylum (Figs 5A and S6). The conservation of the −10-motif and its associated transcription processes (Figs 5B and S8A–F) are consistent with those found in *Deinococcus* (Figs 1E and F and S3A–F). Moreover, the upstream sequences of ORFs containing the −10-motif from these species are fully functional within *D. radiodurans*, suggesting a uniform role of the −10-motif as a promoter within the *D-T* phylum (Fig 5C and D). Beyond the −10-motif, classical gene expression paradigms also exist within the *D-T* phylum (Fig S15).

There is speculation that leaderless genes represent an ancient mode of gene expression (Zheng et al, 2011; Beck & Moll 2018), and bacteria within the *D-T* phylum are considered to be ancient (Zheng et al, 2011; Coleman et al, 2021). Besides, many species within the *D-T* phylum are extremophiles, exhibiting heat, desiccation, and radiation resistance. Therefore, whether the −10-motif expression pattern is simply a remnant of an ancient expression mode in the course of evolution or contributes to the survival in the extreme environment remains a question.

In summary, species within the *D-T* phylum, exemplified by *D. radiodurans*, possess a substantial number of genes that use the −10-motif as the −10 region of a promoter for the expression of leaderless mRNA through RNAP recruitment (Fig 5D). A sequence with a −10-motif at the proper site is sufficient to ensure gene transcription in the *D-T* phylum. And the distinct preference of −10 sequence and −35 sequence among RNAPs from different species is worth further study.

# Materials and Methods

### Bacterial genomes and sequence analysis

Initially, the complete reference genome file (GCF_020546685.1) of *D. radiodurans* (Repar et al, 2021) was downloaded from NCBI on 30 March 2023. A 20-bp DNA sequence upstream of each annotated ORF was extracted and encapsulated into a FASTA file. MEME software (Bailey & Elkan 1994; Bailey et al, 2015) was then used to scan for potential hidden motifs within these sequences (command: meme upORFseqs.txt -dna -w 6 -mod zoops -nmotifs 2 -brief 50,000 -p 1). The scan results are shown in Table S1. The same

procedure was followed for *D-T* phylum species listed in Table S2 (command: meme upORFseqs.txt -dna -w 10 -mod zoops -nmotifs 2 -brief 50,000 -p 1).

Furthermore, for the identification of the potential −10-motif, the 450-bp sequence starting from 300 bp upstream of each ORF in *D. radiodurans* was extracted, and FIMO software (Bailey et al, 2015) was used to scan for possible −10-motifs (command: fimo --thresh 0.015 --norc meme_out/meme.html upORFseqs_300_150.txt). Suitable sequence information for statistical analysis was then filtered out, and the filtered results are shown in Table S3. Amino acid sequences of all ORFs in *D. radiodurans* and ORFs carrying −10-motif were further clustered using EggNOG (Huerta-Cepas et al, 2019), with detailed results shown in Tables S4 and S5, respectively.

For motif scanning in the bacterial domain, reference genomes of species with complete sequencing were downloaded from NCBI on 10 December 2024, and species that were not clearly classified or symbiotic were excluded. The species information used is listed in Table S6. Motif scanning was then conducted using MEME software (command: meme upORFseqs.txt -dna -w 10 -mod zoops -nmotifs 3 -brief 50,000 -p 1) for 20-bp upstream ORFs (results are shown in Table S7). The clustering analysis of high-confidence motifs was performed using the motifStack package in R (Ou et al, 2018).

## Strains, culture media, primers, and reagents

The *E. coli* strains used in the study were *E. coli* Trans5a and *E. coli* BL21(DE3), both acquired from TransGen Biotech Co., Ltd. The strains used (*D. radiodurans* R1 and *T. thermophilus* HB8) were both sourced from the laboratory stock. Primers and ssDNA substrates used in the study (Table S8) were synthesized by Tsingke Biotech Co., Ltd. Unless indicated otherwise, all chemical reagents used in the study were obtained from Sangon Biotech (Shanghai) Co., Ltd. The culture medium for *E. coli* was Luria–Bertani (LB) medium (1% tryptone, 1% NaCl, 0.5% yeast extract), with solid media supplemented with 1.5% agar. The final concentrations of kanamycin and ampicillin used were 40 and 50 μg/ml, respectively. For *D. radiodurans* and *T. thermophilus*, the culture medium was TGY medium (0.5% tryptone, 0.1% glucose, 0.3% yeast extract), with solid media supplemented with 1.5% agar. The final concentration of chloramphenicol in the antibiotic selection medium for *D. radiodurans* was set at 4 μg/ml.

## The construction of the vectors

The plasmids used in this study are listed in Table S9. The PCR system employed in the article was Tks Gflex DNA Polymerase from Takara Biomedical Technology (Beijing) Co., Ltd. For the construction of the expression vector 28a-eCFP, the pET-28a(+) plasmid was linearized using primers pet-f and pet-r, followed by the linearization of *ecfp* using primers fp-pet-f and fp-pet-r. Subsequently, linearized plasmid and *ecfp* were treated with Exnase II (Nanjing Vazyme Biotech Co., Ltd.) together and then transformed into *E. coli* Trans5a, thereby completing the construction of 28a-eCFP. For 28a-rpoDc-eCFP, the construction was

further enhanced by connecting a truncated rpoD fragment to the 28a-eCFP. For the construction of plasmid pK-groES::mCherry, the plasmid pRADK was first linearized using primers pragdk-fp-r and pgroes-fp-r, and the *mCherry* was amplified using primers fp-f and fp-r. After treatment of the linearized plasmid and *mCherry* with Exnase II together, the product was transformed into *E. coli* Trans5a, thereby completing the plasmid construction. For pK-pprI::eGFP, the plasmid pRADK was first linearized using primers pragdk-f and pragdk-fp-r, followed by the amplification of the *pprI* upstream sequence (the upstream sequence refers to the sequence between this ORF and the previous one) using primers pppri-pr-f and pppri-eg-r, and the amplification of the *eGFP* gene using primers egfp-f and fp-r. The three linearized sequences were then treated with Exnase MultiS (Nanjing Vazyme Biotech Co., Ltd.) together and then transformed into *E. coli* Trans5a, thereby completing the plasmid construction. For pK-gC-pI::eG, the obtained plasmid pK-pprI::eGFP was first linearized using primers pragdk-f and pppri-mc-f, followed by the extraction of *groes::mCherry* sequence from pK-groES::mCherry using primers pgroes-pk-f and fp-r. The above method was then employed to obtain the desired plasmid. For pK-gC-mE::eG and pK-gC-dO::eG, pK-gC-pI::eG was linearized using mCherry-r and egfp-f, with the *mqnE* upstream sequence and *ddrO* upstream sequence amplified using the respective primers. Following the same procedure, the desired plasmids were obtained. For other derivative plasmids, the corresponding plasmids were linearized using the corresponding primers, and were then directly transformed it into *E. coli* Trans5a to complete the construction.

## The preparation and transformation of *D. radiodurans*–competent cells

The *D. radiodurans* R1 strain was inoculated into the TGY culture medium and cultured at 30°C with shaking at 200 rpm (Stab S2 Stackable Incubator Shaker, Radobio Scientific Co., Ltd.) until the optical density ($OD_{600}$) reached ~1.0. 1 ml of the bacterial solution was centrifuged at 6,000 rpm (Centrifuge 5424 R, Rotor FA-45-24-11, Eppendorf) for 3 min to remove the supernatant. The cells were resuspended after adding 250 μl of 4 × TGY and 250 μl of 60 mM $CaCl_2$ solution and centrifuged to remove the supernatant. After repeating the resuspension and centrifugation steps once more, 250 μl of 4 × TGY and 250 μl of 60 mM $CaCl_2$ solution were added to resuspend the cells, and the cells were incubated at 30°C with shaking for 90 min to complete the preparation of *D. radiodurans*–competent cells. Subsequently, 500 μl of 50% glycerol was added, and the cells were aliquoted into 1.5-ml EP tubes at 100 μl per tube and stored at −80°C for future use. For transformation, the prepared *D. radiodurans*–competent cells were thawed on ice, and then, 5 μl of plasmid (~100 ng/μl) was added. The mixture was incubated on ice for 30 min, followed by the addition of 900 μl of TGY culture medium and further incubation at 30°C with shaking at 200 rpm (Stab S2 Stackable Incubator Shaker, Radobio Scientific Co., Ltd.) for 6 h. After incubation, the cells were centrifuged at 8,000 rpm (Centrifuge 5424 R, Rotor FA-45-24-11, Eppendorf) for 1 min, leaving ~200 μl of the supernatant. The cell suspension was then spread onto an antibiotic-containing plate, which was then

inverted and incubated in a 30°C incubator (LRH-250 Incubator, Shanghai Yiheng Technology Instrument Co., Ltd.). Once colonies reached the appropriate size, they were selected and cultured in TGY liquid medium.

## Expression and purification of proteins

The recombinant plasmids 28a-eCFP and 28a-rpoDc-eCFP were transformed into *E. coli* BL21(DE3). After verification of protein expression, the expressing strain was cultured in 500 ml of LB medium at 37°C with shaking at 200 rpm (Stab S2 Stackable Incubator Shaker, Radobio Scientific Co., Ltd.). This was continued until the optical density at 600 nm ($OD_{600}$) reached ~0.6. IPTG was then added at a final concentration of 0.2 mM, and the culture was incubated at 30°C with shaking for an additional 6 h to induce protein expression. After the induction, the cells were harvested by centrifugation at 8,000 rpm (Centrifuge 5910 R, Rotor FA-6x250, Eppendorf) for 5 min and washed with PBS. The cell pellet was stored at −80°C for later use. For protein purification, the cells were resuspended in 30 ml of buffer A (1 M NaCl, 5% glycerol, 20 mM Tris–HCl, pH 7.5) and lysed using a cell homogenizer for 40 min, and the lysate was centrifuged to collect the supernatant. The supernatant was then filtered through a 0.22-$\mu$m filter. Protein purification was carried out using an AKTA pure 25 system (GE Healthcare) equipped with a nickel column. The system was first equilibrated with buffer A, after which the filtered supernatant was loaded onto the nickel column. The column was washed with 10% buffer B (1 M NaCl, 5% glycerol, 20 mM Tris–HCl, pH 7.5, 500 mM imidazole) to remove nonspecifically bound proteins, followed by a wash with 40% buffer B to elute the target protein. A final wash with 100% buffer B was performed to clean the column. The eluted protein was then subjected to buffer exchange into buffer C (250 mM NaCl, 5% glycerol, 20 mM Tris–HCl, pH 7.5) using a desalting column. The purified protein was stored at 4°C for short-term use or frozen in liquid nitrogen and stored at −80°C.

## Fluorescence microscopy observation and flow cytometry counting

The *D. radiodurans* R1 strain receiving plasmid transformation was inoculated into a 5 ml TGY culture medium and cultured with shaking until the $OD_{600}$ reached ~5.0. 20 $\mu$l of the bacterial culture was stained with DAPI for 5 min and then photographed from the red (filter cube: Texas Red, excitation filter: 560/40, dichroic mirror: 595, barrier filter: 630/60), green (filter cube: FITC, excitation filter: 480/30, dichroic mirror: 505, barrier filter: 535/45), and blue (filter cube: DAPI, excitation filter: 362/396, dichroic mirror: 415, barrier filter: 432/482) channels using 100× oil objective of ECLIPSE Ti2-U (Nikon). In addition, 1 ml bacterial culture was analyzed for eGFP expression using a CytoFLEX flow cytometer (Beckman Coulter, Inc.). Fluorescence intensity below 800 is considered nonfluorescent, and if fewer than 10 out of every 10,000 cells express fluorescence, it is considered that there is no fluorescence expression. *E. coli* Trans5a harboring the target plasmid was cultured

overnight in LB medium, followed by observation and photography under the fluorescence microscope.

## Quantitative real-time PCR (qRT–PCR)

The corresponding *D. radiodurans* strains were inoculated into 50 ml of TGY culture medium and cultured with shaking until the $OD_{600}$ reached ~1.0, after which the cells were centrifuged and preserved. Total RNA was extracted using TransZol Up Plus RNA Kit (TransGen Biotech), and reverse transcription was carried out on 1 $\mu$g of RNA using Reverse Transcriptase M-MLV (Takara). Quantitative real-time PCR was subsequently conducted using TB Green Premix Ex Taq (Takara) and the Mx3005P fluorescence quantitative PCR instrument (Stratagene), following a two-step protocol. The gene of interest was *eGFP* (primers: egfp-rt-f 5′-CGGCAAGCTGACCCT GAAGT-3′ and egfp-rt-r 5′-TTCATGTGGTCGGGGTAGCGG-3′), whereas the *mCherry* was used for normalization (primers: mc-rt-f 5′-CAG GACTCCTCCCTGCAGGA-3′ and mc-rt-r 5′-GCGTCGTAGTGGCCGCCGTC-3′).

## FRET assays

The 5′-FAM–labeled ssDNA was prepared in a 10 $\mu$M solution with double-distilled water ($ddH_2O$), whereas the purified eCFP and RpoDc-eCFP were formulated in a 10 $\mu$M solution with buffer C. The ssDNA and proteins were added in a 1:1 volume ratio, and the resulting reaction mixture was incubated at 37°C for 30 min. After incubation, the sample was then transferred to a black-bottom, opaque 384-well plate (Corning), and the emission spectra from 460 to 600 nm were recorded using the FlexStation 3 multimode microplate reader (Molecular Devices) with an excitation wavelength of 440 nm. FRET efficiency is obtained by calculating the ratio of emission intensities at 515 and 480 nm. Relative FRET efficiency is obtained by calculating the ratio of FRET efficiency of the same ssDNA incubated with RpoDc-eCFP and eCFP, respectively.

## Transcriptome sequencing and transcript quantity statistics

The *T. thermophilus* HB8 strain was cultured in 50 ml of TGY medium at 70°C in a water bath incubator (THZ-82A Water Bath Incubator Shaker, Changzhou Jintan Liangyou Instrument Co., Ltd.), with shaking at 220 rpm until the $OD_{600}$ reached 1.0. The cells were then pelleted by centrifugation at 8,000 rpm (Centrifuge 5910 R, Rotor FA-6x250, Eppendorf) for 5 min. The harvested bacteria were sent to Novogene Co., Ltd. for transcriptome sequencing. Briefly, sequencing libraries were generated using NEBNext Ultra RNA Library Prep Kit for Illumina (NEB), and library quality was assessed on the Agilent 5400 system (Agilent). Qualified libraries were pooled and sequenced on Illumina platforms with a PE150 strategy. Subsequently, the transcriptome of *D. radiodurans* (SRX2731648, WT stain without treatment) and *T. thermophilus* (SRR28238428) was processed for library construction and sequence alignment using the makeblastdb tool. For quantifying transcript levels of target genes, the ORF and 400-bp regions upstream and downstream of the ORF were selected. The target sequence was aligned with the established sequence

alignment libraries using the BLAST tool, and results containing gaps or inversions were excluded. Transcript levels for each nucleotide were calculated based on the alignment data, and the statistical results of the ORF and 100-bp regions upstream and downstream of the ORF were documented.

### Structure prediction and molecular dynamics simulation

Truncated RpoD proteins from *D. radiodurans* (WP_227086025.1, Met168-Ile311), *T. thermophilus* (WP_011228005.1, Tyr84-Ile258), and *E. coli* (WP_000437376.1, Thr95-Ile450) that interact with the –10 region (–10-motif) were selected for structure prediction. Using the DNA strand 5′-TGTTAAGCTCCC-3′ as a template, the AlphaFold server (Abramson et al, 2024) was employed to predict their structures. After obtaining the predicted structures, the phosphate group at the 5′ end of the DNA strand was removed for molecular dynamics simulations. For the molecular dynamics simulation, the size of the solution system was set to $8 \times 10 \times 11$ nm$^3$. Sodium ions and chloride ions were added to reach a salt concentration of 125 mM. The simulations were performed using GROMACS 2024.3 (Abraham et al, 2015) with the AMBER14SB_ParmBSC1 force field (Maier et al, 2015; Ivani et al, 2016) and TIP3P water molecules (Jorgensen et al, 1983). After initial equilibration, all systems were simulated at 310 K and 1 atm for 50 ns with a time step of 2 fs. The molecular dynamics simulations were repeated three times, and the binding free energy ($\Delta G$) between protein and DNA was calculated using the gmx_MMPBSA method (Valdes-Tresanco et al, 2021). The root mean square deviation and energies were recorded every 0.02 ns during the simulation process.

### Statistical analysis and reproducibility

The predicted protein structures presented in the article were sourced from the AlphaFold DB, and the protein crystal structures were obtained from the PDB. Because of the absence of a predicted structure for RpoD from *D. radiodurans*, its structure was predicted using the AlphaFold2 prediction server (Jumper et al, 2021). Structural alignments were conducted using PyMOL v2.6.0a0. DNA sequence diagrams were prepared using SnapGene v6.0.2, and statistical graphs were created with Origin 2019b. All experimental results were successfully replicated at least three times. Unless indicated otherwise, the results represented the means and SD of independent experiments. To compare significant differences, a one-way ANOVA followed by Tukey's post hoc test was conducted.

## Data Availability

The transcriptome data of *T. thermophilus* are available in NCBI (SRR28238428). The code used for motif scanning and transcription quantification can be obtained from GitHub (https://github.com/Zpresitong/upORF-motif-discover.git). The data that support the findings of this study are available from the corresponding author upon reasonable request.

## Supplementary Information

## Acknowledgements

This research was funded by the Zhejiang Provincial Natural Science Foundation of China (LQ23C010002, LZYQ25C010002, and LGN22C010002) and the National Natural Science Foundation of China (32200016 and 32370028).

### Author Contributions

S Zhong: software, investigation, methodology, and writing—original draft.
Ln Wang: investigation, methodology, and writing—original draft.
S Song: investigation, methodology, and writing—review and editing.
La Wang: funding acquisition, investigation, and writing—review and editing.
Y Hua: conceptualization, funding acquisition, and writing—review and editing.
H Lu: conceptualization, funding acquisition, and writing—review and editing.

### Conflict of Interest Statement

The authors declare that they have no conflict of interest.

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
