## [Reviewer comments · Life Science Alliance]

Life Science Alliance

The -10 region adjacent to open reading frames is a common expression pattern in *Deinococcus-Thermus*

Shitong Zhong, Linjia Wang, Shuang Song, Liangyan Wang, Yuejin Hua, and Huizhi Lu

DOI: <https://doi.org/10.26508/lsa.202503354>

Corresponding author(s): Huizhi Lu, Zhejiang University and Yuejin Hua, Zhejiang University

Review Timeline:

Submission Date:	2025-04-09
Editorial Decision:	2025-06-18
Revision Received:	2025-07-04
Editorial Decision:	2025-07-25
Revision Received:	2025-07-30
Accepted:	2025-07-31

Scientific Editor: Sarita Hebbar

Transaction Report:

June 16, 2025

Re: Life Science Alliance manuscript #LSA-2025-03354-T

Huizhi Lu

MOE Key Laboratory of Biosystems Homeostasis & Protection, Institute of Biophysics, College of Life Sciences, Zhejiang University, China

Dear Dr. Lu,

Thank you for submitting your manuscript entitled "The -10 region adjacent to open reading frames is a common gene expression pattern in *Deinococcus-Thermus*" to Life Science Alliance. The manuscript was assessed by two expert reviewers, whose comments are appended to this letter.

Both reviewers have sought details in the rationale, experimental design, and in the methods section. We agree that the manuscript needs to be revised to include all the requested details and use of specific terminology as suggested by the reviewers.

In line with the overall recommendations, we invite you to submit a revised manuscript addressing all the reviewers' comments. When submitting the revision, please include a letter addressing the reviewers' comments point by point. While a rebuttal must respond to all points in some form, additional data to resolve these points may not be required.

Thank you for this interesting contribution to Life Science Alliance. We are looking forward to receiving your revised manuscript.

Sincerely,

Sarita Hebbar, PhD
Scientific Editor
Life Science Alliance
<http://www.lsjournal.org>

-- By submitting a revision, you attest that you are aware of our payment policies found here: <https://www.life-science->

B. MANUSCRIPT ORGANIZATION AND FORMATTING:

Reviewer #1 (Comments to the Authors (Required)):

The manuscript provides experimental evidence for distinct -10 promoter elements in the D-T phylum. Overall the experiments significantly advance the field and provide a basis for moving away from the commonly held paradigm of prokaryotic transcription which is heavily biased towards E. coli. I have the following comments:

- While I think the manuscript and experiments are very good, I think the authors need to reorganise the manuscript a little to really highlight their work in the context of the literature. I think the authors also need to better highlight that current knowledge of transcription is currently based on knowledge from E. coli, and clearly highlight how their study shows differences to these models.

- An introduction with more work on transcription in D-T is needed. I found several important papers that were not highlighted in the introduction.

- An absence of 5' UTR should be explained before the term 'leaderless' is used. It took me some time to properly understand what was meant by the term 'leaderless'

- Deinococcus-Thermus phylum appears inaccurate is also Deinococcota which should be mentioned.

- How were the genomes used in the analysis selected? There are several hundred Deinococcota genomes available in NCBI and I would think that a bioinformatic analysis of promoters would benefit to a large-scale widespread analysis. Could the authors please explain why there are only limited number of genomes used in the study?

- The genomes used in the study should be listed in a supporting table instead of in the text.

- Line 252. Please also provide the strain designation for each organism.

- Line 334. Please be aware that the bioinformatic detection of the -10 and RBS sequences are only predictions and not the confirmed RBS of these organisms. Experimental evidence is needed to make this claim. Therefore please rewrite the sentence to indicate it is prediction and not fact.

- Line375. Please provide more specific information. If the -10 motif location differs from others, state the common distances of the -10 motif in other organisms.

- Line 457. When making statements such as 'existing studies have shown that the optimal distance between the -35 region and the -10 region is 17{plus minus}1 bp' please be sure to make this organism specific. I would also like to make this comment for the rest of the manuscript as well.

- An absence of 5' UTR should be explained before the term 'leaderless' is used. It took me some time to properly understand what was meant by the term 'leaderless'

- Deinococcus-Thermus phylum appears inaccurate is also Deinococcota which should be mentioned.

- How were the genomes used in the analysis selected? There are several hundred Deinococcota genomes available in NCBI and I would think that a bioinformatic analysis of promoters would benefit to a large-scale widespread analysis. Could the authors please explain why there are only limited number of genomes used in the study?
- The genomes used in the study should be listed in a supporting table instead of in the text.
- Line 252. Please also provide the strain designation for each organism.
- Line 334. Please be aware that the bioinformatic detection of the -10 and RBS sequences are only predictions and not the confirmed RBS of these organisms. Experimental evidence is needed to make this claim. Therefore please rewrite the sentence to indicate it is prediction and not fact.
- Line 375. Please provide more specific information. If the -10 motif location differs from others, state the common distances of the -10 motif in other organisms.
- Line 457. When making statements such as 'existing studies have shown that the optimal distance between the -35 region and the -10 region is 17{plus minus}1 bp' please be sure to make this organism specific. I would also like to make this comment for the rest of the manuscript as well.

Reviewer #2 (Comments to the Authors (Required)):

The authors explore the DNA motifs upstream of *D. radiodurans* genes and their effect on gene expression, a quest interesting in the light of findings that *D. radiodurans* abounds with leaderless genes. Having found two statistically significant motifs within 20 bp upstream of *D. radiodurans* genes, the authors compared them to the classical promoter structure motifs. They found that the sequence of the first resembled the classical -10-motif and of the second the classical ribosome binding site (RBS) motif. However, the positioning of the found -10-motif (the *D. radiodurans* -10-motif in further text) didn't follow the classical paradigm; it was too close to the translation start site of genes; in fact, close enough to be in a space usually reserved for the RBS sequence and to mark leaderless genes. The authors then proceeded to characterize the function of this motif, confirming that its function corresponds to the function of the classical -10-motif.

Specifically, the authors looked closely at the function of the *D. radiodurans* -10-motif and found (i) by looking at the transcriptome data that it associated with a localized increase in transcription, similar to the promoters of classical configuration, which also do appear in *D. radiodurans*, (ii) the *D. radiodurans* -10-motif, as well as the classical -10-motif, cannot be replaced with random sequences without loss of gene expression, (iii) the *D. radiodurans* motif cannot be replaced with an RBS motif without loss of gene expression, (iv) through FRET experiments that the *D. radiodurans* Rpo interacts specifically with the *D. radiodurans* -10-motif, (v) that removing the upstream 50 bp containing the *D. radiodurans* -10-motif reduces gene expression, (vi) that expression is restored if the sequence at the -10 position of this construct is replaced with the *D. radiodurans* -10-motif (mostly, depending somewhat on the surrounding sequences), (vii) that adding the -35 sequence at 17bp, but not 22bp, distance from the *D. radiodurans* -10-motif enhances gene expression, mirroring the conformation of the transcriptional apparatus, (viii) that the *D. radiodurans* -10-motif is a common occurrence within the *Deinococcus-Thermus*, (ix) by looking at the transcriptome data of *T. thermophilus* that, similarly to *D. radiodurans*, it associates with a localized increase in gene expression of specific genes, and (x) that the motifs from different *Deinococcus-Thermus* species increase gene expression in *D. radiodurans*.

Overall, the design of experiments is convincing in figuring out the exact function of the *D. radiodurans* -10 motif. I have some questions/suggestions:

I suggest the authors choose different names than the -10-motif and -10-region to distinguish between the *D. radiodurans* and the classical -10-motif, as these names are too generic - e.g. doesn't the -10-region of the classical promoter structure contain a motif that can, consequently, be thought of as a -10-motif?

It is presumed that the genes with the *D. radiodurans* -10-motif are leaderless. Has this been previously shown, or is it the finding of the authors, or both?

Line 366 Regarding the SRX2731648 dataset - please explain in the Methods which transcripts were used - the wt or the mazEF mutant, without or with the MMC treatment. Is the expression of the genes chosen for Figure 1 linked to any treatment? How were these genes chosen? Has expression been detected for all the genes containing the *D. radiodurans* -10-sequence?

Minor comments:

Introduction - The *Deinococcus-Thermus* phylum has recently been renamed (into *Deinococcota*), you can mention this in the Introduction.

Line 80 - delete „Of"

Line 83 - rephrase: „has enabled bioinformatics analyses that sequences upstream of" into "has enabled bioinformatics analyses showing that sequences upstream of"

Line 84 - „reveal" instead of „reveals"

Lines 121 - 134 Please cite relevant papers for the genome assemblies where appropriate e.g. *Deinococcus radiodurans* - Repar et al 2021, <https://doi.org/10.1038/s41598-021-89173-9>, *Deinococcus geothermalis* - Makarova et al 2007, <https://doi.org/10.1371/journal.pone.0000955>, *Meiothermus silvanus* - Complete genome sequence of *Meiothermus silvanus* type strain (VI-R2), <https://doi.org/10.4056/sigs.1042812>

Line 338 - unclear section title - different than what? Please rephrase.

Line 344 - wouldn't it be logical to cut of the leading NN from the putative *D. radiodurans* RBS sequence „NNGGAG" if NN presents any two nucleotides?

Line 345-5 - „The RBS-motif aligns well with the RBS sequence in both sequence and position," Do the authors mean the classical RBS sequence? The authors should better linguistically distinguish between the *D. radiodurans* RBS motif and the classical RBS motif, throughout the paper.

Line 358-360 Sentence unclear. Are the authors trying to say that the *pprI* gene was previously annotated as leaderless?

Line 364 "To determine whether there is any regulation preference of -10-motifs versus classical promoters..." Don't the classical promoters also contain the -10-motifs?? The authors should better linguistically distinguish between the *D. radiodurans* -10-motif and the canonical -10-motif, throughout the paper.

Figure 2 - which ORF start position is used for the *PprI* gene? The previously annotated one, or the new one?

Figure 3- D,E - line 761-2: what are p*PprI* point mutations, are they also ssDNAs

445-449 sentence scrambled, please rephrase.

Line 477 "demonstrating the universal of -10-motif in D-T phylum." Do the authors mean the universality?

Line 487 "researchers have identified leaderless mRNAs that lacking or possessing very short 5'-UTRs, which can still be normally" - delete the word "that"

Line 497-8 "This motif, functioning in transcription, was used as a marker to identify the leaderless gene."

Line 515. Were the added regions -10 and -35 from the *E. coli*, or was the -10 region from *D. radiodurans*?

Discussion: My understanding is that there is an overarching question about leaderless genes on what influences differential expression of specific sets of these genes within a cell e.g. upon change of external conditions. Perhaps the authors can add a section on this in the discussion, in light of their findings that the sequence around the *D. radiodurans* -10-motif has some influence on the intensity of expression (as shown in Figure 4B).

Response to reviewers:

We would like to thank you for your careful reading, helpful comments, and constructive suggestions. We have carefully considered all comments and suggestions from the reviewers and revised our manuscript accordingly. In the following sections, we summarized our responses to your comments. We hope that our responses have well addressed your concerns.

Reviewer #1 (Comments to the Authors (Required)):

The manuscript provides experimental evidence for distinct -10 promoter elements in the D-T phylum. Overall the experiments significantly advance the field and provide a basis for moving away from the commonly held paradigm of prokaryotic transcription which is heavily biased towards *E. coli*. I have the following comments:

Authors' response: Thank you for your thorough review of our manuscript. We carefully reviewed your comments and suggestions and made thorough revisions to the manuscript based on them.

- While I think the manuscript and experiments are very good, I think the authors need to reorganise the manuscript a little to really highlight their work in the context of the literature. I think the authors also need to better highlight that current knowledge of transcription is currently based on knowledge from *E. coli*, and clearly highlight how their study shows differences to these models.

Authors' response: Thank you for your recognition of our work. In the introduction section, we simply highlighted that existing knowledge is based on *E. coli* studies and highlighted the differences in our findings within the *D-T* phylum:

“In the classical expression model primarily studied in *E. coli*, gene transcription relies on promoters containing -35 and -10 regions to produce mRNA with 5'-UTR; and subsequent translation depends on the RBS located within this 5'-UTR. In contrast, our research reveals that a significant number of genes in the *D-T* phylum utilize transcription relying on a -10 region immediately upstream of the ORF to

produce leaderless mRNA for protein expression. This pattern is markedly different from the classical expression model.” (Line 121-127, Page 5)

- An introduction with more work on transcription in D-T fanyis needed. I found several important papers that were not highlighted in the introduction.

Authors’ response: Thank you for your suggestion. We have incorporated related research literature (Mekler et al., PMID: 23087380; Miropolskaya et al., PMID: 22605342; Weixlbaumer et al., PMID: 23374340; Bae et al., PMID: 26349032; Miropolskaya et al., PMID: 29097207) into the introduction section. (Line 93-96, Page 4)

- An absence of 5' UTR should be explained before the term 'leaderless' is used. It took me some time to properly understand what was meant by the term 'leaderless'

Authors’ response: Thank you for your helpful suggestions. We have added an explanation of “leaderless” (“mRNA products do not have or only have a very short 5'-untranslated region”) in the manuscript to ensure its meaning is more clearly conveyed. (Line 29-30, Page 2)

- Deinococcus-Thermus phylum appears inaccurate is also Deinococcota which should be mentioned.

Authors’ response: It is indeed a point that requires attention. We have included the nomenclature of “*Deinococcota*” in our manuscript. (Line 5, Page 2; Line 468, Page 17)

- How were the genomes used in the analysis selected? There are several hundred Deinococcota genomes available in NCBI and I would think that a bioinformatic analysis of promoters would benefit to a large-scale widespread analysis. Could the authors please explain why there are only limited number of genomes used in the study?

Authors’ response: Thank you for your question. In fact, we have analyzed all

- The genomes used in the study should be listed in a supporting table instead of in the text.

Authors' response: Thank you for your suggestion. We have included the genomes we used in a supplementary table (Table S2, part of the table is shown below.).

Assembly Accession	Organism Name
GCF_000091545.1	Thermus thermophilus HB8
GCF_020546685.1	Deinococcus radiodurans R1 = ATCC 13939 = DSM 20539
GCF_000024425.1	Meiothermus ruber DSM 1279
GCF_003444775.1	Deinococcus ficus
GCF_000196275.1	Deinococcus geothermalis DSM 11300
GCF_000800395.1	Deinococcus radiopugnans
GCF_011067105.1	Deinococcus wulumuqiensis R12
GCF_001507665.1	Deinococcus actinosclerus
GCF_020889625.1	Deinococcus radiophilus
GCF_000092425.1	Truepera radiovictrix DSM 17093
GCF_000252445.1	Deinococcus gobiensis I-0
GCF_000020685.1	Deinococcus deserti VCD115
GCF_001535545.1	Thermus parvatiensis
GCF_000317835.1	Deinococcus peraridilitoris DSM 19664
GCF_000186385.1	Deinococcus maricopensis DSM 21211
GCF_000092125.1	Allomeiothermus silvanus DSM 9946 (Meiothermus silvanus)
GCF_000183745.1	Oceanithermus profundus DSM 14977
GCF_000195335.1	Marinithermus hydrothermalis DSM 14884
GCF_000190555.1	Deinococcus proteolyticus MRP
GCF_045784305.1	Deinococcus radiomollis
GCF_025997855.1	Deinococcus aetherius
GCF_003173015.1	Deinococcus irradiatoli
GCF_045784255.1	Deinococcus altitudinis
GCF_025244745.1	Deinococcus rubellus
GCF_040256395.1	Deinococcus sonorensis KR-87
GCF_001644565.1	Deinococcus puniceus
GCF_020229735.1	Deinococcus taeanensis
GCF_003860465.1	Deinococcus psychrotolerans
GCF_004684245.1	Thermus caldilimi
GCF_001399775.1	Thermus aquaticus Y51MC23
GCF_003568585.1	Meiothermus taiwanensis WR-220
GCF_000309885.1	Thermus oshimai JL-2
GCF_022846375.1	Thermus brockianus
GCF_018771645.1	Thermus antranikianii
GCF_046774955.1	Deinococcus yunweiensis
GCF_028622095.1	Deinococcus aquaticus

Assembly Accession	Organism Name
GCF_004758605.1	Deinococcus metallilatus
GCF_000381045.1	Thermus scotoductus DSM 8553
GCF_001485435.1	Deinococcus grandis
GCF_000771745.2	Thermus filiformis
GCF_014201875.1	Deinococcus humi
GCF_900102145.1	Thermus arciformis
GCF_021462405.1	Thermus tengchongensis
GCF_007990775.1	Deinococcus cellulosityticus NBRC 106333 = KACC 11606
GCF_007990995.1	Meiothermus granaticius NBRC 107808
GCF_007991015.1	Oceanithermus desulfurans NBRC 100063
GCF_000701405.1	Deinococcus marmoris DSM 12784
GCF_007990975.1	Meiothermus hypogaeus NBRC 106114
GCF_030804085.1	Deinococcus enclensis
GCF_900176165.1	Deinococcus hopiensis KR-140
GCF_000373205.1	Calidithermus timidus DSM 17022
GCF_000421625.1	Thermus islandicus DSM 21543
GCF_000381345.1	Deinococcus apachensis DSM 19763
GCF_000482805.1	Deinococcus murrayi DSM 11303
GCF_000378445.1	Deinococcus aquatilis DSM 23025
GCF_000376265.1	Thermus igniterrae ATCC 700962
GCF_000430045.1	Calidithermus chliarophilus DSM 9957
GCF_000423425.1	Meiothermus rufus DSM 22234
GCF_000519345.1	Deinococcus pimensis DSM 21231
GCF_002901445.1	Deinococcus koreensis
GCF_009687825.1	Deinococcus kurensis
GCF_003590815.1	Deinococcus cavernae
GCF_000744885.1	Thermus amyloliquefaciens
GCF_003426945.1	Thermus sediminis
GCF_042661385.1	Deinococcus lacus
GCF_039545155.1	Deinococcus aluminii
GCF_042649445.1	Deinococcus rufus
GCF_042648985.1	Deinococcus metalli
GCF_009377345.1	Deinococcus terrestris
GCF_022760855.1	Thermus albus
GCF_014202645.1	Deinobacterium chartae
GCF_014648135.1	Deinococcus sedimenti
GCF_014646915.1	Deinococcus aquiradiocola
GCF_014201885.1	Deinococcus budaensis
GCF_002897375.1	Deinococcus aerius
GCF_014647995.1	Deinococcus arenae
GCF_003217515.1	Deinococcus yavapaiensis KR-236
GCF_014647435.1	Deinococcus radiotolerans

Assembly Accession	Organism Name
GCF_003028415.1	Deinococcus arcticus
GCF_003336745.1	Thermus caldifontis
GCF_019693445.1	Deinococcus aquaedulcis
GCF_014647655.1	Deinococcus malanensis
GCF_039522025.1	Deinococcus depolymerans
GCF_014647175.1	Deinococcus daejeonensis
GCF_014647055.1	Deinococcus aerolatus
GCF_039545055.1	Deinococcus carri
GCF_004307015.1	Thermus thermamylovorans
GCF_009982895.1	Deinococcus alpinitundrae
GCF_018863415.1	Deinococcus aestuarii
GCF_020166415.1	Deinococcus multiflagellatus
GCF_042656085.1	Deinococcus petrolearius
GCF_014648155.1	Deinococcus knuensis
GCF_900109185.1	Deinococcus reticulitermitis
GCF_042431605.1	Deinococcus oregonensis
GCF_014646895.1	Deinococcus roseus
GCF_007280555.1	Deinococcus detaillensis
GCF_020166395.1	Deinococcus betulae
GCF_014648115.1	Deinococcus seoulensis
GCF_009755355.1	Deinococcus arboris
GCF_004634215.1	Deinococcus fonticola
GCF_002869765.1	Deinococcus planocerae
GCF_022760925.1	Thermus hydrothermalis
GCF_022760955.1	Thermus neutrinimicus
GCF_000745915.1	Deinococcus misasensis DSM 22328
GCF_000620065.1	Meiothermus cereboreus DSM 11376
GCF_014648095.1	Deinococcus ruber
GCF_014647075.1	Deinococcus aerophilus
GCF_019430925.1	Thermus brevis
GCF_000599865.1	Deinococcus phoenicis
GCF_021462525.1	Thermus caliditerrae
GCF_014647535.1	Thermus composti
GCF_039544935.1	Deinococcus caeni
GCF_022760835.1	Thermus altitudinis
GCF_042685025.1	Deinococcus taklimakanensis
GCF_000701425.1	Deinococcus frigens DSM 12807
GCF_033088415.1	Deinococcus arenicola
GCF_022760915.1	Thermus thalpopphilus
GCF_002964845.1	Thermus tenuipunicus
GCF_009834985.1	Deinococcus xianganensis
GCF_003574095.1	Calidithermus roseus

Assembly Accession	Organism Name
GCF_042649355.1	Deinococcus antarcticus
GCF_003574085.1	Meiothermus luteus
GCF_003574345.1	Calidithermus terrae
GCF_030271215.1	Deinococcus rhizophilus
GCF_042653785.1	Deinococcus hohokamensis
GCF_042653905.1	Deinococcus navajonensis
GCF_002198095.1	Deinococcus indicus
GCF_014653275.1	Deinococcus piscis
GCF_040361885.1	Deinococcus xinjiangensis
GCF_965137195.1	Deinococcus saxicola

- Line 252. Please also provide the strain designation for each organism.

Authors' response: We have added information about the strain types in the manuscript such as *D. radiodurans* R1 and *E. coli* Trans5a. (Line 248, 256, Page 10)

- Line 334. Please be aware that the bioinformatic detection of the -10 and RBS sequences are only predictions and not the confirmed RBS of these organisms. Experimental evidence is needed to make this claim. Therefore please rewrite the sentence to indicate it is prediction and not fact.

Authors' response: Thank you for your suggestion. We have rewritten the sentences in the manuscript to make their meanings clearer:

“These two predicted motifs, although similar in sequence to known functional sequences, still require further validation for their specific functions. Since the RBS-motif not only shares similar features in sequence with classical RBS sequences but also appears a few bp upstream of the ORF, matching the positional characteristics of classical RBS” (Line 339-343, Page 13)

- Line375. Please provide more specific information. If the -10 motif location differs from others, state the common distances of the -10 motif in other organisms.

Authors' response: Thank you very much for your suggestion. Generally, in other species, the ORF is always directly followed by the RBS upstream. Therefore, it can

be inferred that the -10 region (or -10-motif) of these species would appear further upstream from the ORF. However, in *D. radiodurans*, a relatively significant portion of the ORF upstream is directly connected to the -10 motif, resulting in a different positioning relative to the ORF. We have added relevant explanations in the manuscript:

“Previous sequence analysis has shown that many species have their ORFs closely followed by RBSs upstream (Omotajo et al. 2015; Nakagawa et al. 2017), and therefore it can be inferred that the classical promoter -10 regions for these ORFs should be located dozens bp upstream.” (Line 373-376, Page 14)

- Line 457. When making statements such as 'existing studies have shown that the optimal distance between the -35 region and the -10 region is 17{plus minus}1 bp' please be sure to make this organism specific. I would also like to make this comment for the rest of the manuscript as well.

Authors' response: Thank you for your suggestion. We have added relevant species information in the manuscript. (Line 463, Page 17)

Reviewer #2 (Comments to the Authors (Required)):

The authors explore the DNA motifs upstream of *D. radiodurans* genes and their effect on gene expression, a quest interesting in the light of findings that *D. radiodurans* abounds with leaderless genes. Having found two statistically significant motifs within 20 bp upstream of *D. radiodurans* genes, the authors compared them to the classical promoter structure motifs. They found that the sequence of the first resembled the classical -10-motif and of the second the classical ribosome binding site (RBS) motif. However, the positioning of the found -10-motif (the *D. radiodurans* -10-motif in further text) didn't follow the classical paradigm; it was too close to the translation start site of genes; in fact, close enough to be in a space usually reserved

for the RBS sequence and to mark leaderless genes. The authors then proceeded to characterize the function of this motif, confirming that its function corresponds to the function of the classical -10-motif.

Specifically, the authors looked closely at the function of the *D. radiodurans* -10-motif and found (i) by looking at the transcriptome data that it associated with a localized increase in transcription, similar to the promoters of classical configuration, which also do appear in *D. radiodurans*, (ii) the *D. radiodurans* -10-motif, as well as the classical -10-motif, cannot be replaced with random sequences without loss of gene expression, (iii) the *D. radiodurans* motif cannot be replaced with an RBS motif without loss of gene expression, (iv) through FRET experiments that the *D. radiodurans* Rpo interacts specifically with the *D. radiodurans* -10-motif, (v) that removing the upstream 50 bp containing the *D. radiodurans* -10-motif reduces gene expression, (vi) that expression is restored if the sequence at the -10 position of this construct is replaced with the *D. radiodurans* -10-motif (mostly, depending somewhat on the surrounding sequences), (vii) that adding the -35 sequence at 17bp, but not 22bp, distance from the *D. radiodurans* -10-motif enhances gene expression, mirroring the conformation of the transcriptional apparatus, (viii) that the *D. radiodurans* -10-motif is a common occurrence within the Deinococcus-Thermus, (ix) by looking at the transcriptome data of *T. thermophilus* that, similarly to *D. radiodurans*, it associates with a localized increase in gene expression of specific genes, and (x) that the motifs from different Deinococcus-Thermus species increase gene expression in *D. radiodurans*.

Overall, the design of experiments is convincing in figuring out the exact function of the *D. radiodurans* -10 motif. I have some questions/suggestions:

Authors' response: Thank you for your careful review of the manuscript and for your recognition of our work. Your summary of the manuscript content is very accurate, and we have carefully revised the manuscript based on your comments and suggestions.

I suggest the authors choose different names than the -10-motif and -10-region to distinguish between the *D. radiodurans* and the classical -10-motif, as these names are too generic - e.g. doesn't the -10-region of the classical promoter structure contain a motif that can, consequently, be thought of as a -10-motif?

Authors' response: Thank you very much for your suggestion.

We had previously considered using entirely different names to refer to the -10-motif, but experimental evidence has shown that except for the lack of -35 region and the generated leaderless transcripts, the -10-motif and the -10 region are not fundamentally different from each other.

The reason we use the term "-10-motif" is to draw a distinction for readers between this concept and the classical -10 region of promoters, but we do not directly refer to "-10-motif" as "-10 region". In essence, we have made a distinction between them. The "-10-motif" places greater emphasis on sequence form, and any sequence in the form of TANNNT can be considered as a -10-motif (in the article, we mainly focus on the characteristic sequences adjacent to the ORF). On the other hand, the "-10 region" is actively performing promoter functions, so we use similar but slightly different terms to reflect the connection between sequences while making a distinction. In terms of sequence form, the -10 region can be regarded as a -10-motif; however, functionally speaking, a sequence in the form of a -10-motif does not necessarily serve as a -10 region in terms of its function—for example, it could occur coincidentally within an ORF. Simply put, the -10-motif emphasizes sequence form, while the -10 region emphasizes actual function. Thank you again for your suggestion.

It is presumed that the genes with the *D. radiodurans* -10-motif are leaderless. Has this been previously shown, or is it the finding of the authors, or both?

Authors' response: Thank you for your question. Previous studies (Zheng et al. 2011, PMID: 21749696; de Groot et al. 2014, PMID: 24723731) have discovered the -10-motif and inferred that the corresponding genes are leaderless. We have further

demonstrated that the -10-motif upstream of the ORF functions as a -10 region, and this pattern is prevalent within the *D-T* phylum.

Line 366 Regarding the SRX2731648 dataset - please explain in the Methods which transcripts were used - the wt or the mazEF mutant, without or with the MMC treatment. Is the expression of the genes chosen for Figure 1 linked to any treatment? How were these genes chosen? Has expression been detected for all the genes containing the *D. radiodurans* -10-sequence?

Authors' response: Thank you for your question. The transcriptome we selected is wild-type and untreated with MMC, and the graphs presented all represent untreated samples. This has been supplemented in the methods section. In addition, *groES* and *ddrO* are genes with very noticeable classic promoter sequence expressions, particularly evident in the significant increase in downstream transcription levels. Regarding the -10-motif, *mqnE* maintains its -10-motif expression pattern within the D-T group (Figure S9), while *pprI* stands out due to annotation errors (Figure S2E). Therefore, these two genes are selected as notable examples. Of course, we also conducted statistics on the transcription levels of sequences near the -10-motif in *D. radiodurans* (Figure S3E, F, shown below), and it can be observed that most of these sequences show an upward trend, which does not contradict our conclusion.

Additionally, we performed the same statistics for *T. thermophilus* as well (Figure S7E, F, shown below), yielding consistent results.

Minor comments:

Introduction - The *Deinococcus-Thermus* phylum has recently been renamed (into *Deinococcota*), you can mention this in the Introduction.

Authors' response: Thank you for your suggestion. We mentioned the new name *Deinococcota* in the introduction. (Line 5, Page 2)

Line 80 - delete "of"

Authors' response: The "of" has been deleted. (Line 81, Page 4)

Line 83 - rephrase: "has enabled bioinformatics analyses that sequences upstream of" into "has enabled bioinformatics analyses showing that sequences upstream of"

Authors' response: The "showing" has been added based on your suggestion. (Line 84, Page 4)

Line 84 - "reveal" instead of "reveals"

Authors' response: The "reveals" has been corrected to "reveal". (Line 85, Page 4)

Lines 121 - 134 Please cite relevant papers for the genome assemblies where appropriate e.g. *Deinococcus radiodurans* - Repar et al 2021, <https://doi.org/10.1038/s41598-021-89173-9>, *Deinococcus geothermalis* - Makarova et al 2007, <https://doi.org/10.1371/journal.pone.0000955>, *Meiothermus silvanus* -

Complete genome sequence of *Meiothermus silvanus* type strain (VI-R2), <https://doi.org/10.4056/sigs.1042812>

Authors' response: Thank you for your suggestion. We have added the genomic sequencing literature of *Deinococcus radiodurans* to the article (Line 132, Page 6), and the genomic information for the remaining species has been attached in a supplementary table (Table S2).

Line 338 - unclear section title - different than what? Please rephrase.

Authors' response: Thank you for your suggestion. We have changed the title as “-10-motif forms a nonclassical expression pattern in *D. radiodurans*” to make the meaning clearer. (Line 333, Page 13)

Line 344 - wouldn't it be logical to cut off the leading NN from the putative *D. radiodurans* RBS sequence "NNGGAG" if NN presents any two nucleotides?

Authors' response: We have deleted these two nucleotides "NN". (Line 338, Page 13)

Line 345-5 - „The RBS-motif aligns well with the RBS sequence in both sequence and position," Do the authors mean the classical RBS sequence? The authors should better linguistically distinguish between the *D. radiodurans* RBS motif and the classical RBS motif, throughout the paper.

Authors' response: Thank you for your suggestion. We have made a distinction between RBS-motif and classical RBS to ensure clear expression. (Line 341, 343, Page 13; and throughout the paper)

Line 358-360 Sentence unclear. Are the authors trying to say that the *pprI* gene was previously annotated as leaderless?

Authors' response: Previous study has indeed discussed annotation errors of *pprI*. To clarify the meaning, we revised the sentences accordingly. (Line 355-356, Page 13)

Line 364 "To determine whether there is any regulation preference of -10-motifs versus classical promoters..." Don't the classical promoters also contain the -10-motifs?? The authors should better linguistically distinguish between the *D. radiodurans* -10-motif and the canonical -10-motif, throughout the paper.

Authors' response: Thank you for your suggestions. In our previous answers, we explained that the "-10-motif" highlights the sequence form, while the "-10 region" is focused on actual function. Actually, the -10 region in the classical promoters is equivalent to -10-motifs adjacent to ORF based on the sequence and function, and we have distinguished between -10 motifs and classical -10 regions in our text. (Line 362, Page 14; and throughout the paper)

Figure 2 - which ORF start position is used for the *PprI* gene? The previously annotated one, or the new one?

Authors' response: The ORF start site of the *pprI* gene we used is new, and has been mentioned in the text. (Figure 2, Line 761, Page 28)

Figure 3- D,E - line 761-2: what are p*PprI* point mutations, are they also ssDNAs

Authors' response: We're sorry for any inconvenience this may have caused. These point mutations are all ssDNAs derived from ssDNA ss-pP, and we have made correction in the text. (Figure 3, Line 782, Page 29)

445-449 sentence scrambled, please rephrase.

Authors' response: Thank you for your suggestion. We have rewritten the sentences to make their meanings clearer. (Line 448-455, Page 16-17)

Line 477 "demonstrating the universal of -10-motif in D-T phylum." Do the authors mean the universality?

Authors' response: Yes, we have changed "universal" to "universality". (Line 482, Page 18)

Line 487 "researchers have identified leaderless mRNAs that lacking or possessing very short 5'-UTRs, which can still be normally" - delete the word "that"

Authors' response: We have deleted "that". (Line 493, Page 19)

Line 497-8 "This motif, functioning in transcription, was used as a marker to identify the leaderless gene."

Authors' response: Thank you for pointing that out. This sentence was repetitive with previous sentences, so we have reorganized it to make the expression clearer. (Line 501, Page 19)

Line 515. Were the added regions -10 and -35 from the *E. coli*, or was the -10 region from *D. radiodurans*?

Authors' response: Thank you for your question. In *E. coli*, all the plasmids expressed were those shown in Figures 2 and 3, so all these -10 region sequences originated from *D. radiodurans*. As for the -35 region sequence, it was the classical TTGACA, which is a result of previous summaries of the promoter sequences in *E. coli*, and this sequence also appears in *D. radiodurans* (e.g., the -35 region sequence of *groES*, shown in Figure 2). (Line 513-518, Page 19-20)

Discussion: My understanding is that there is an overarching question about leaderless genes on what influences differential expression of specific sets of these genes within a cell e.g. upon change of external conditions. Perhaps the authors can add a section on this in the discussion, in light of their findings that the sequence around the *D. radiodurans* -10-motif has some influence on the intensity of expression (as shown in Figure 4B).

Authors' response: Thank you for your suggestion. During our experiments, we indeed observed that -10-motifs in different sequences exhibit varying expression strengths. We speculate that this might be influenced by the adjacent sequences. Of course, binding motifs of some transcription factors could also appear near -10-motifs, thereby regulating expression strength, such as the binding site of DdrO (Figure S13,

shown below). We have supplemented relevant discussions in our paper. (Line 522-527, Page 20)

ddrA	GCGTT TTATGCTTGACCGTAA TGTTATTCTGTTCTAAACTAAATGC ATG	TTATGCTTGACCGTAA	DdrO binding site
ddrC	GTGCTGATTG TTATGTCAAAAACATAA TCTGTGCTAGAAATATCTGT ATG	TAAACT	-10-motif
ruvB	CAACCCAAGCTGTTTCCTTGAT TTTCGCAAATAGCGTAA TATGCATCC ATG	ATG	Start codon
mutS	CCTGCC TTTCGCTCAGAACGTAA AGACCTGCGCTATCTTGTCCCT ATG		
recQ	GGGCCTCCATGAT TTCTGCCACACGTAA ACGCGCTATCCTGGGGAG ATG		

July 25, 2025

RE: Life Science Alliance Manuscript #LSA-2025-03354-TR

Dr. Huizhi Lu
Zhejiang University
Yuhangtang Road 886
Hangzhou, Zhejiang 310058
China

Dear Dr. Lu,

Thank you for submitting your revised manuscript entitled "The -10 region adjacent to open reading frames is a common expression pattern in *Deinococcus-Thermus*". Your revised manuscript was evaluated by both the original reviewers. In line with their comments, we would be happy to publish your paper in Life Science Alliance pending final revisions necessary to meet our formatting guidelines.

- Kindly pay attention to the minor suggestion from Reviewer 2. In line with this, we recommend that you conduct a thorough spell and grammar check of the entire manuscript document. If you use AI-based grammar/spelling checkers please note this in the methods section.
- Please complete the fluorescence microscopy section of the methods to include information on the objective (magnification, numerical aperture), the excitation and emission filters used.
- Please provide the sequence details for primers used in the qRT-PCR experiment.
- Please indicate clearly in the "Data Availability" section if data included in the manuscript will be deposited in any public repository.
- Please add scale bars for figures 2C, 3J, 4B, 5C, S4B, S5A, B and S11A.
- Please consult our manuscript preparation guidelines <https://www.life-science-alliance.org/manuscript-prep> and make sure your manuscript sections are in the correct order.
- Please add your main, supplementary figure, and table legends to the main manuscript text after the references section.
- Please remove figure legends from the figures. Legends should be provided only in the manuscript file.
- Please carefully go through all your figure legends and revise them such that they contain sufficient information to allow the reader to follow the data presented without referring back to the text.
- Please revise legend for Figure S14 such that the figure panels are introduced in alphabetical order.
- Please add callouts for Figures 1F; S2A-D; S3A-F; S4A-B; S8A-F; S11A-B; S12A-E and S14A-C to your main manuscript text.
- Please upload all figure files as individual ones, including the supplementary figure files; all figure legends should only appear in the main manuscript file.
- Please add ORCID ID for corresponding (and secondary corresponding) author--you should have received instructions on how to do so
- Please add a Summary Blurb/Alternate Abstract and a Category in our system.
- Please add the X and Bluesky handles of your host institute/organisation, as well as your own and/or one of the authors, in our system.
- Please be sure that the authorship listing and order is correct.

A. FINAL FILES:

B. MANUSCRIPT ORGANIZATION AND FORMATTING:

Sincerely,

Sarita Hebbar, PhD
Scientific Editor
Life Science Alliance
<http://www.lsajournal.org>

Reviewer #1 (Comments to the Authors (Required)):

The manuscript is much improved and I commend the authors on their positive response and implementation of all reviewer comments.

Reviewer #2 (Comments to the Authors (Required)):

Generally, I am satisfied with the authors' response. There are still some minor writing issues, e. g. in the Abstract: Line 29. „leaderless genes (mRNA products do not have or only have a very short 5'-untranslated region)" Add „that" to the sentence („mRNA products THAT do not have...")

Responses:

-Kindly pay attention to the minor suggestion from Reviewer 2. In line with this, we recommend that you conduct a thorough spell and grammar check of the entire manuscript document. If you use AI-based grammar/spelling checkers please note this in the methods section.

Authors' response: Thank you for your reminder. We have made revisions based on Reviewer 2's comments and have also conducted a thorough check of the entire manuscript for any spelling or grammatical errors.

-Please complete the fluorescence microscopy section of the methods to include information on the objective (magnification, numerical aperture), the excitation and emission filters used.

Authors' response: We have included these contents in the revised manuscript. "20 μ L of the bacterial culture was stained with DAPI for 5 minutes, and then photographed from the red (filter cube: Texas Red, excitation filter: 560/40, dichroic mirror: 595, barrier filter: 630/60), green (filter cube: FITC, excitation filter: 480/30, dichroic mirror: 505, barrier filter: 535/45), and blue (filter cube: DAPI, excitation filter: 362/396, dichroic mirror: 415, barrier filter: 432/482) channels by using 100 \times oil objective of ECLIPSE TI2-U (Nikon, Japan)."

-Please provide the sequence details for primers used in the qRT-PCR experiment.

Authors' response: We have included the sequences of the primers used in the qRT-PCR experiment in the revised manuscript.

-Please indicate clearly in the "Data Availability" section if data included in the manuscript will be deposited in any public repository.

Authors' response: We have added this section to the revised manuscript.

-Please add scale bars for figures 2C, 3J, 4B, 5C, S4B, S5A, B and S11A.

Authors' response: Scale bars were added as requested.

-Please consult our manuscript preparation guidelines <https://www.life-science-alliance.org/manuscript-prep> and make sure your manuscript sections are in the correct order.

Authors' response: The order of the manuscript sections has been arranged as requested.

-Please add your main, supplementary figure, and table legends to the main manuscript text after the references section.

Authors' response: We have added legends to the main manuscript text.

-Please remove figure legends from the figures. Legends should be provided only in the manuscript file.

Authors' response: The figures do not contain any legends.

-Please carefully go through all your figure legends and revise them such that they contain sufficient information to allow the reader to follow the data presented without referring back to the text.

Authors' response: We have checked each figure's legend, ensuring that their meanings are clearly expressed.

-Please revise legend for Figure S14 such that the figure panels are introduced in alphabetical order.

Authors' response: We have revised the legend as requested.

-Please add callouts for Figures 1F; S2A-D; S3A-F; S4A-B; S8A-F; S11A-B; S12A-E and S14A-C to your main manuscript text.

Authors' response: We have added the callouts as requested.

-Please upload all figure files as individual ones, including the supplementary figure files; all figure legends should only appear in the main manuscript file.

Authors' response: All figure files are uploaded as requested.

-Please add ORCID ID for corresponding (and secondary corresponding) author--you should have received instructions on how to do so

Authors' response: We have added ORCID ID for the corresponding author.

-Please add a Summary Blurb/Alternate Abstract and a Category in our system.

Authors' response: A Summary Blurb and a Category have been added.

-Please add the X and Bluesky handles of your host institute/organisation, as well as your own and/or one of the authors, in our system.

Authors' response: These social networking sites are not accessible from China.

-Please be sure that the authorship listing and order is correct.

Authors' response: Confirmed.

Authors' response: We appreciate your clear guidance on the necessary revisions.

Reviewer #1 (Comments to the Authors (Required)):

The manuscript is much improved and I commend the authors on their positive response and implementation of all reviewer comments.

Authors' response: We appreciate your careful review and positive evaluation of our work.

Reviewer #2 (Comments to the Authors (Required)):

Generally, I am satisfied with the authors' response. There are still some minor writing

issues, e. g. in the Abstract:

Line 29. „leaderless genes (mRNA products do not have or only have a very short 5'-untranslated region)"

Add „that" to the sentence („mRNA products THAT do not have...")

Authors' response: We appreciate your valuable feedback and careful review of our work. We have corrected the errors pointed out and conducted a thorough check on spelling and grammar throughout the entire manuscript.

July 31, 2025

RE: Life Science Alliance Manuscript #LSA-2025-03354-TRR

Dr. Huizhi Lu
Zhejiang University
Yuhangtang Road 886
Hangzhou, Zhejiang 310058
China

Dear Dr. Lu,

Thank you for submitting your Research Article entitled "The -10 region adjacent to open reading frames is a common expression pattern in *Deinococcus-Thermus*". It is a pleasure to let you know that your manuscript is now accepted for publication in Life Science Alliance. Congratulations on this interesting work.

Your manuscript will now progress through copyediting and proofing. A minor request is to consider editing the alternate abstract for accuracy during the proofing stage.

It is journal policy that authors provide original data upon request. The final published version of your manuscript will be deposited by us to PubMed Central upon publication.

DISTRIBUTION OF MATERIALS:

Again, congratulations on a very nice paper. I hope you found the review process to be constructive and are pleased with how the manuscript was handled editorially. We look forward to future exciting submissions from your lab.

Sincerely,

Sarita Hebbar, PhD
Scientific Editor
Life Science Alliance
<http://www.lsajournal.org>